# Zero-Shot Adaptation of Behavioral Foundation Models to Unseen Dynamics

**Maksim Bobrin**[1,2]    **Ilya Zisman**[5]
[1]AXXX    [2]Applied AI Institute, Computational Imaging Lab    [5]Humanoid

**Alexander Nikulin**[1,3]    **Vladislav Kurenkov**[3,4]    **Dmitry Dylov**[1,2]
[1]AXXX    [2]Applied AI Institute, Computational Imaging Lab    [3]MSU    [4]Innopolis University

## ABSTRACT

Behavioral Foundation Models (BFMs) have proven successful at producing near-optimal policies for arbitrary tasks in a zero-shot manner, requiring no test-time retraining or task-specific fine-tuning. Among the most promising BFMs are those that estimate the successor measure, learned in an unsupervised way from task-agnostic offline data. However, these methods fail to react to changes in the dynamics, making them ineffective under partial observability or when the transition function changes. This hinders the applicability of BFMs in real-world settings (e.g., robotics), where the dynamics can change unexpectedly at test time. In this work, we demonstrate that the Forward–Backward (FB) representation, cannot produce reasonable policies under distinct dynamics, leading to interference among latent policy representations. To address this, we propose an FB model with a transformer-based belief estimator, which greatly facilitates zero-shot adaptation. Additionally, we show that partitioning the policy-encoding space into dynamics-specific clusters aligned with context-embedding directions yields further performance gains. These traits allow our method to respond to the dynamics mismatches observed during training and to generalize to unseen dynamics changes. Empirically, in settings with changing dynamics, our approach achieves up to 2× higher zero-shot returns than baselines on both discrete and continuous tasks.

Project page: https://belief-fb.dunnolab.ai/

## 1 INTRODUCTION

One very desirable property of reinforcement learning (RL) agents is their ability to adapt during inference to new tasks without requiring any additional learning. Achieving this in as few trials as possible would be even better: the ideal being the zero-shot adaptation (Barreto et al., 2017; Touati et al., 2022), where the agent never interacts with the environment at test-time and relies solely on the task-agnostic data. Behavioral Foundational Models (BFMs) (Pirotta et al., 2023; Sikchi et al., 2024; Tirinzoni et al.) may be considered as a step in this direction, because they can learn a variety of behaviors from offline data without task specific labels. During inference, it is possible to extract a task-specific policy that is theoretically optimal in terms of performance. Recent works (Tirinzoni et al.) demonstrates that methods based on *successor measure* estimation through Forward-Backward (FB) decomposition (Touati & Ollivier, 2021), is especially versatile and can successfully extract diverse policies from data.

At the same time, FB has a fundamental drawback that limits its adaptation ability to more general settings. In our paper, we show that FB is unable to generalize across different environment configurations (*i.e.*, dynamics), such as changes in a transition function (*e.g.*, new obstacles or environment configuration) or some latent factor variation (*e.g.*, wind direction or change in mass). This limitation stems from the way the approximation of the *successor measure* (Blier et al., 2021) is estimated: FB averages the discounted future-occupancy state distribution over all observed dynamics, which inevitably causes *interference* in a policy representation space. This fact alone may severely constrain the applicability of FB in the real-world scenarios. For example, one of the largest robotics dataset, Open X-Embodiment (Collaboration, 2023), consists of 22 different robot embodiments, and training

---

corresponding author: maxs.bobrin@gmail.com
work is done in collaboration with dunnolab.ai

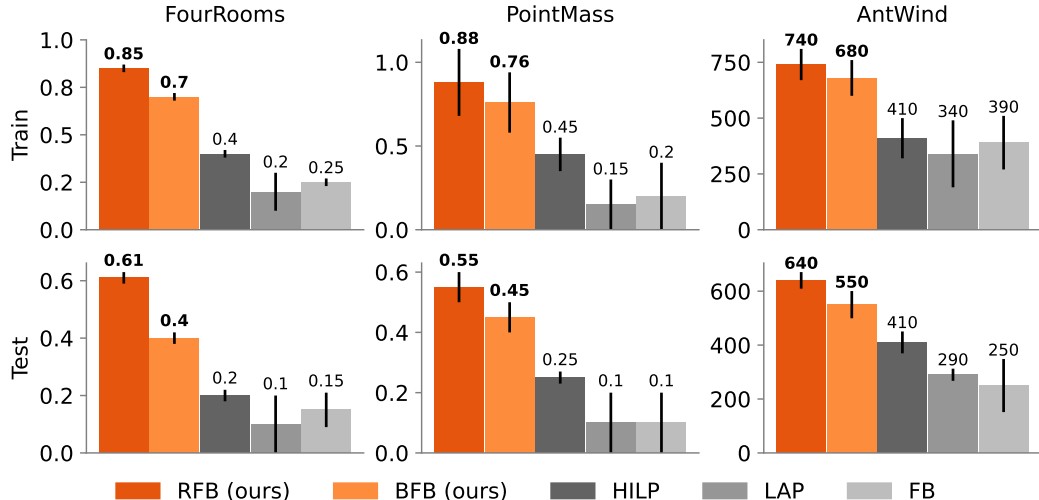

Figure 1: **Summary of results**. Aggregate mean performance over *seen* (train) and *unseen* (test) dynamics for zero-shot RL. The error bars indicate standard deviation over 5 seeds. Notably, both BFB and RFB adapt not only to the dynamics seen during training, but are also able to generalize to unseen dynamics. There are $30(20)$ training (test) dynamics for Randomized-FourRooms and PointMass and $16(4)$ for AntWind environments.

FB on each of them independently is infeasible. In Section 3.1, we discuss the underlying cause in more details by providing both empirical and theoretical support.

To remedy such limitation, we introduce **Belief-FB (BFB)**, a conditioning method for FB through a *belief* estimation, a popular technique of uncertainty quantification in Meta-RL (Zintgraf et al., 2020; Dorfman et al., 2021). To implement this, we employ a permutation-invariant transformer encoder, denoted as $f_{\text{dyn}}$, which processes a given trajectory from the dataset to produce a dynamics-encoding vector $h$. This representation is subsequently utilized as a conditioning input to the successor feature network, expressed as $F(\cdot, \cdot, h, \cdot)$. We pre-train $f_{\text{dyn}}$ in a self-supervised fashion, thus posing no additional requirements on the data structure or the trajectory re-labeling, while maintaining theoretical guarantees. We discuss the implementation of Belief-FB in Section 3.2.

Remarkably, Belief-FB enables the generalization capabilities of FB not only through the dynamics seen in the training dataset, but also on the *unseen test* configurations. Additionally, we find that in order to align *belief* estimation better with FB, one also needs to partition the policy space encodings prior into dynamics-specific clusters, which significantly improves generalization abilities of FB. We propose **Rotation-FB (RFB)** that accomplishes this partitioning. We present the theoretical support and the implementation details of Rotation-FB in Section 3.3. Empirically, both BFB and RFB outperform baselines for seen and unseen dynamics, as gathered in Figure 1 and discussed in Section 4.3.

We believe that our work sufficiently broadens the possible applicability of BFMs, yet keeping all of the zero-shot properties unchanged. Our contributions are as follows:

- **We demonstrate the limitation of Forward-Backward (FB) representations** (Touati & Ollivier, 2021), which lies in its inability to generalize *per se* across different dynamics both from train and test, where dynamics shift constitute of new layout grids or changes in the transition function that are hidden from an agent. Refer to Section 3.1 for more discussion.

- **We propose Belief–FB (BFB)**, which employs a permutation-invariant transformer encoder to infer a belief over the current dynamics (Zintgraf et al., 2020; Dorfman et al., 2021). Analyzing BFB's policy encoding space reveals that additional disentanglement is beneficial, motivating our Rotation–FB (RFB) extension. Section 3.2 examines Belief-FB, and Section 3.3 details Rotation-FB's theoretical motivation and implementation.

- **We empirically demonstrate that both BFB and RFB can adapt to different dynamics**, unlike its counterparts in the zero-shot setup. Refer to Section 4.3 for the discussion and Figure 1 for results.

## 2 BEHAVIORAL FOUNDATION MODELS (BFMS)

A Behavioral Foundation Model (BFM) (Pirotta et al., 2023; Tirinzoni et al.; Frans et al., 2024; Park et al., 2024; Sikchi et al., 2025) is an RL agent trained in an unsupervised manner on a task-agnostic dataset to approximate optimal policies for diverse reward functions (tasks) specified at inference (test-time).

*Forward-Backward Representation (FB)* (Touati & Ollivier, 2021) approximates a discounted successor measure (Blier et al., 2021) for various behaviors across diverse tasks. The successor measure $M^\pi(s_0, a_0, X)$ for subset $X \subset \mathcal{S}$ is defined as cumulative discounted time spend at $X$ starting at $(s_0, a_0)$ and following $\pi$ thereafter, defined as (for finite state-action space case)

$$M^\pi(s_0, a_0, X) = \sum_{t \geq 0} \gamma^t \mathrm{P}(s_{t+1} \in X | s_0, a_0, \pi) \tag{1}$$

with the corresponding Q-function for a specific task $r$:

$$Q_r^\pi(s_0, a_0) = \sum_{s^+ \in X} r(s^+) M^\pi(s_0, a_0, s^+). \tag{2}$$

In the continuous case, the FB representation aims to approximate successor measure through finite-rank approximation under diverse policies through *forward* $F : \mathcal{S} \times \mathcal{A} \times \mathcal{Z} \to \mathbb{R}^d$ and *backward* $B : \mathcal{S} \to \mathbb{R}^d$ functions. Given a set of policies $\pi_z$ parametrized by task variable drawn uniformly from sphere $z_{\mathrm{FB}} \in \mathrm{Unif}(\mathcal{Z} = \mathbb{S}^d)$. Assuming $\rho$ is a probability distribution over states within the offline dataset, the objective for FB is written as $M^{\pi_z}(s_0, a_0, X) \approx \int_{s^+ \in X} F(s_0, a_0, z)^T B(s^+) \rho(ds)$. Then, policy can be extracted as :

$$\pi_z(s) \approx \arg\max_a F(s, a, z)^T z. \tag{3}$$

For continuous case, the greedy policy is approximated via DDPG (Lillicrap et al., 2015). We refer to the Appendix B.1 for in-depth details regarding FB training procedure. During test time the task policy parametrization is approximated as $z_{test} \approx \mathbb{E}_{(s,a) \sim \rho}[r_{test}(s, a) B(s, a)]$. If the inferred task vector $z_{test}$ lies within the task sampling distribution (in a linear span) of $\mathcal{Z}$ used during training, then the optimal policy for task $r_{test}$ is obtained from Equation 2 as $\pi_z(s) \approx \arg\max_a Q_{r_{test}}^{\pi_z}(s, a)$. Extended discussion on other related works is included in the Appendix A.

## 3 METHOD

**Problem Statement.** The dataset consisting of diverse environments can be formally considered as a Contextual Markov Decision Process (CMDP) defined by a context space $\mathcal{C}$ and a mapping $\mathcal{M} : c \in \mathcal{C} \mapsto \mathcal{M}(c) = (\mathcal{S}, \mathcal{A}, T_c, r_c, \rho_c, \gamma)$, where both $\mathcal{S}, \mathcal{A}$ are shared across contexts, $T_c : \mathcal{S} \times \mathcal{A} \to \Delta(\mathcal{S})$ is the context-dependent transition kernel, $r_c$ is the reward function, $\rho_c \in \Delta(\mathcal{S})$ is the initial state distribution, and $\gamma \in [0, 1)$ is the discount factor. Each context $c$ (e.g., wind direction, friction, or door locations) specifies a unique MDP.

When the context $c$ is unobserved, the problem becomes a POMDP. Under standard assumptions, there exists a sufficient history-dependent statistic—the *belief state* $b_t(c) = \mathbb{P}(c \mid H_t) \in \Delta(\mathcal{C})$—capturing the posterior over contexts given the history $H_t$. Solving the POMDP is equivalent to solving the fully observable *Belief-MDP* $(\Delta(\mathcal{C}) \times \mathcal{S}, \mathcal{A}, T_b, r_b, \rho_b, \gamma)$, where states are augmented with beliefs.

We assume access to an offline, reward-free dataset $\mathcal{D}_{\mathrm{train}}$ consisting of trajectories $\{(s_k, a_k, s_{k+1})\}_{k=1}^N$ collected under diverse exploratory policies from a finite set of training contexts $C_{\mathrm{train}} \subset \mathcal{C}$. At test time, for an unseen context $c_{test} \in \mathcal{C} \setminus C_{\mathrm{train}}$, we are given a short reward-free history $H = \{(s_t, a_t, s_{t+1})\}_{t=0}^L$ from an exploratory policy in $\mathcal{M}(c_{test})$, and a task specified by reward $r : \mathcal{S} \times \mathcal{A} \to \mathbb{R}$.

The goal is to infer an approximate belief $\hat{b}(c \mid H)$ for any given history (trajectory) and extract a zero-shot policy $\pi$ (without additional learning) that minimizes the regret

$$\mathcal{R} = \sup_{c_{\mathrm{test}} \in \mathcal{C} \setminus C_{\mathrm{train}}, \, r} \mathbb{E}_{(s,a) \sim \rho_{c_{\mathrm{test}}}} \left[ Q_r^{\pi^*}(s, a) - Q_r^\pi(s, a) \right], \tag{4}$$

---

We use the term "zero-shot RL" following Touati & Ollivier (2021).

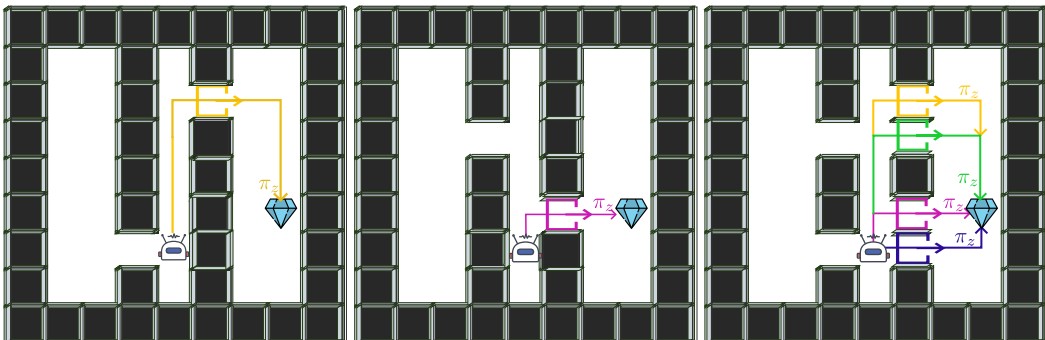

Figure 2: **Randomized-Doors environment for three different layouts, each produced through varying the grid structure (exact randomization procedure is a hidden variable)** (*left-middle*) From state $s$, the goal of an agent is to capture a diamond at target location by picking up the most suitable policy $\pi_z$ (yellow for the first type and purple for the second) to move to the closest open door based on internal representation. (*right*) When there are multiple possible future outcomes in the training data from the same state, the $\pi_z$'s (different colors) interfere with each other, leading to picking up an averaged policy.

where $\pi^*$ is the optimal policy for task $r$ under dynamics $T_{c_{\text{test}}}$. To formally study optimality guarantees of the problem above, we employ the following assumption commonly used for dynamics generalization (Eysenbach et al., 2021; Jeen & Cullen, 2024):

**Assumption 1** (Coverage). The test initial state–action distribution $\rho_{\text{test}}$ is supported on the support of $\rho$, i.e. $\text{supp}(\rho_{\text{test}}) \subseteq \text{supp}(\rho)$ (equivalently, $\rho_{\text{test}} \ll \rho$, i.e absolutely continuous).

## 3.1 INVESTIGATING LATENT DIRECTIONS SPACE UNDER MULTIPLE DYNAMICS

We begin by addressing the following question: *Why does FB representations fail to generalize effectively to different situations under dynamics variations,* i.e., *if learned on data sampled from diverse CMDPs?* While the answer may appear intuitive, a closer look into the geometric structure of learned latent directions $z_{\text{FB}} \in \mathcal{Z}$, which encode possible policies $\pi_z$ reveals critical insights which will be helpful later. We approach this question both theoretically and empirically on custom didactic discrete partially-observable Randomized Doors (see Appendix C.1) environment for building intuition. Partial observability adds additional challenges and showcases the need to estimate belief state, which we discuss in the following sections.

In this experiment the only source of dynamics variation is the grid layout type. Namely, the positions of doors and walls are changed each new episode, depending on hidden configuration variable $c$. We collect a dataset of random trajectories drawn from multiple layouts, yielding near-uniform coverage of the entire state-action space. Now, consider a particular state $s$ that an agent finds itself in three different layouts (see Figure 2). During FB training, we estimate expected successor features via $F(s, \cdot, z_{\text{FB}})$ for policy representations $z_{\text{FB}} \sim \text{Uniform}(\mathbb{S}^{d-1})$ starting at $s$.

In this setting, optimal successor features require different optimal policies, depending on the layout an agent is instantiated in. Because $z_{\text{FB}}$ does not enforce a separation (*i.e.*, prior) over layout-specific futures, the FB model suffers from *interference*: latent directions encoding conflicting future outcomes overlap and become entangled in the policy representation space $\mathcal{Z}$. For each layout configuration and fixed state $s$, Figure 3 depicts latent directions $z_{\text{FB}}$, colored by optimal policy as $a_{\text{color}} = \arg\max_a F(s, a, z_{\text{FB}})^T z_{\text{FB}}$. When FB is trained on first two layouts separately, a unique dominant behavior (colored) emerges in $\mathcal{Z}$, recovering the optimal goal-reaching policy $\pi_z^*$. This means that any randomly sampled $z$ at training time agrees on the possible future (successor features). In contrast, training on mixed data, where transitions come from multiple environment instances, causes $z_{\text{FB}}$ to **blend dynamics-specific information** and **average over possible futures**, yielding a policy that is suboptimal for every layout, including those in the training set. These observations are theoretically supported by the following:

**Theorem 1** (Regret bound via uniform successor approximation under Assumption 1). *Let $r$ be bounded with $\|r\|_\infty \leq R$ with $R = \sup_{(s,a) \in \mathcal{S} \times \mathcal{A}} |r(s,a)|$ and discount $\gamma \in (0,1)$. For any test*

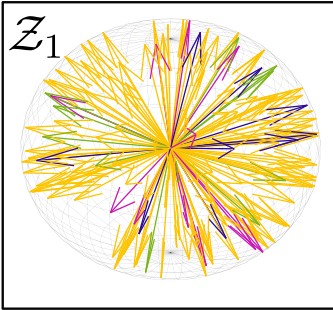 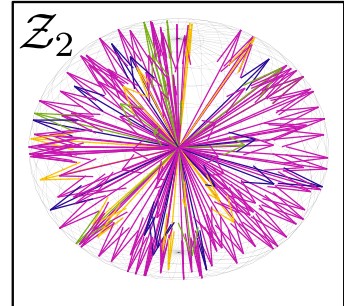 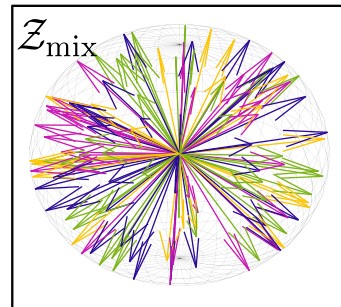

Figure 3: **Visualization of the randomly sampled vectors $z_{\mathbf{FB}}$, which encode policies $\pi_{z_{\mathbf{FB}}}$ for three datasets from [Figure 2](#). For a fixed state $s$ and same goal state across configurations, arrows depict latent directions $z_{\mathbf{FB}} \in \mathcal{Z}$ and colored by dominant directions as** $a_{color} = \arg\max_a F(s, a, z_{\mathbf{FB}})^T z_{\mathbf{FB}}$. (*left-middle*) When FB is trained on the two configurations independently, most of the latent directions agree on the optimal policy $\pi_z$. (*right*) When FB is trained on mix of CMDPs and at test time tasked with any particular configuration from train, obtained policy is ambiguous, since most policy-encoding directions do not agree on the action.

*CMDP satisfying the coverage assumption, the policy $\pi_{\hat{z}}$ returned by the method obeys*

$$\mathbb{E}_{(s,a)\sim\rho_{\text{test}}}\left[Q_r^*(s,a) - Q_r^{\pi_{\hat{z}}}(s,a)\right] \leq \frac{3}{1-\gamma} R\left(\varepsilon_k^* + \Delta_{\text{est}}\right). \tag{5}$$

We provide a full proof in [Appendix B](#). Because $\epsilon_{k+1} \geq \epsilon_k$ by monotonicity of the worst-case approximation error over a fixed function class, the upper bound in equation 5 becomes looser as more environments are included at training time. This statement concerns the approximation term only. In practice, adding CMDPs may also increase the dataset size and reduce the finite-sample estimation term (see additional discussion in the [Appendix B](#)), so the net effect on regret is empirical.

Futher, in [Section 3.3](#), we refine this result and show that the explicit dependence on the total number of environments $k$ can be replaced by a dependence on $k_{\max}$ (the size of the largest cone/cluster), thereby tightening the upper bound when $k_{\max} \ll k$.

This interference highlights a fundamental trade-off. FB is expressive enough to model any task in the linear span of the reward, and yet, when trained across environments with distinct unobserved parameters, the lack of contextual conditioning forces it to average expected future across different dynamics rather than separate them. The resulting successor measure merges transitions from distinct layouts and entangles directions in the latent space $\mathcal{Z}$. To disentangle these directions, we must represent uncertainty about the hidden context explicitly. The next section introduces a belief-conditioned objective that infers the latent context and allows FB to maintain environment-specific successor measures.

> **Takeaway 1**
>
> Because FB training inherently averages over all possible successor features, it cannot learn a disentangled policy space and, therefore, fails to adapt to changes in dynamics.

## 3.2 BELIEF STATE MODELING

To resolve the interference issue described in [Section 3.1](#), we **infer the latent context of an environment and augment FB input on that belief**. We train a transformer encoder $f_{\text{dyn}}$, by passing to a *set* of transitions $\{(s_t, a_t, s'_{t+1})\}_{t=1}^N$ and outputting an $h \in \mathbb{R}^d$. We denote the space of all possible inferred contexts as $\mathcal{H}$, where each element $h$ encodes dynamics for particular environment. Because the ordering is discarded and no rewards in transitions are provided, the encoder must focus on dynamics specific mismatches (*e.g.*, layout geometry, friction or wind direction), rather than policy specifics. Such context encoder should be permutation invariant, since unobservable factors describing environment are independent of the order of transitions in an episode. This setting provides theoretical ground for zero-shot and few-shot learning [Snell et al. (2017)](#).

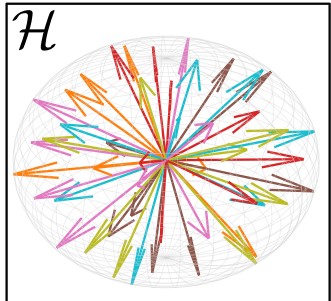 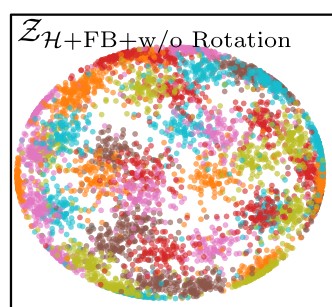 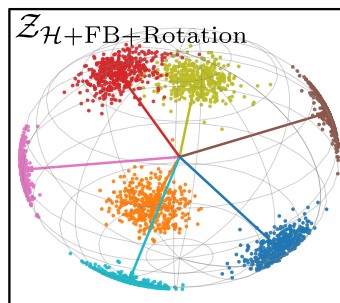

Figure 4: **Inferred context vectors and task vectors across training stages.** Arrows represent inferred contexts $h \in \mathcal{H}$; points on the sphere boundary represent task vectors $z_{\text{FB}}$ (middle panel). Points are colored by CMDP, so transitions from the same CMDP share the same color. The concentration parameter $\kappa$ controls the cluster dispersion. (*left*) Untrained $f_{\text{dyn}}$: inferred contexts are unstructured; transitions from the same CMDP are not aligned. (*middle*) New sampling procedure (before training): $z_{\text{FB}}$ is encouraged to align with the corresponding $h$, but clusters across CMDPs still overlap. (*right*) After training: contexts from the same CMDP form coherent, aligned clusters, and task vectors $z_{\text{FB}}$ separate across environment configurations, reducing interference.

Concretely, dataset consists of episodes $(\{(s_t, a_t, s'_{t+1})_{c_i}\}_{t=1}^N$ coming from CMDPs with randomly instantiated hidden specification variable $c_i$ (different dynamics). We train a transformer encoder on random episodes (without episodic labels $c_i$) of context length $n$ to infer contextual (hidden) variable $h$ which fully specifies the dynamics across given episode. The transformer encoder loss involves two main components: 1) $h$ is encouraged to follow a Gaussian prior and is shared across trajectory, and 2) projection head, which combines $h$ with $(s_{(}t, a_t)$ to predict $s_{t+1}$. Those stages can be either trained end-to-end or separately. We observed that separating FB training from $f_{\text{dyn}}$ gives better results.

For each trajectory we concatenate the inferred context vector $h$ with the task vector $z_{\text{FB}}$ to obtain augmented input $[h; z_{\text{FB}}]$ and condition only forward network as:

$$\hat{M}_{\pi_z}(s_t, a_t, s_{t+1}) = F(s_t, a_t, [h; z_{\text{FB}}])^T B(s_{t+1}). \tag{6}$$

We empirically found that conditioning the backward network $B$ degraded performance, producing smoothed out $Q$ function, so $B$ remains shared across contexts. Algorithm is summarized in Algorithm 1.

At test time, the agent is provided with a short (context length), reward-free trajectory and it is passed to $f_{\text{dyn}}$ to obtain $h$. By plugging the result into Equation 3, the greedy policy is obtained.

> **Takeaway 2**
>
> We train a transformer in a self-supervised regime to estimate a belief over possible contexts, augmenting FB inputs and enabling effective disentanglement of contextual representations.

### 3.3 STRUCTURING DIRECTIONS IN THE LATENT SPACE

Insights from Section 3.1 showed that sampling task-vectors $z_{\text{FB}}$ uniformly on the hypersphere encodes averaged policies, while Section 3.3 provided a solution through explicit context identification. We now combine these observations together through enhanced sampling $z_{\text{FB}}$ around the inferred context $h$.

In Vanilla-FB, each state $s$ draws $z_{\text{FB}} \sim \text{Unif}(\mathbb{S}^{d-1})$ with no inductive bias, so resulting policies $\pi_z$ conflict with each other in CMDP setting, **even if additional explicit conditioning is introduced as before**. We replace uniform prior with a *von Mises-Fisher* (vMF) distribution centered at the context direction for episode $h = f_{\text{dyn}}(\{(s_i, a_i, s_{i+1})\})$ as

$$z_{h+\text{FB}} \sim \text{vMF}(\mu = h, \kappa). \tag{7}$$

with $\kappa$ controlling the spread or *diversity* of policies (left and middle figures from Figure 4). In practice, to draw $z_{h+\text{FB}}$ we first pick a simple vector (*e.g.*, the first basis vector), perturb with vMF noise, and finally rotate the result onto $h$ with Householder reflection.

This enhancement has several benefits: 1) because directions $h$ that differ in dynamics now occupy disjoint cones on the hypersphere, FB can fit the successor measure locally inside each cone, avoiding the destructive averaging effect quantified in Section 3.1 and 2) alignment procedure encourages the agent to explore policies that are plausible under its current belief while still injecting controlled diversity through $\kappa$.

Importantly, such a procedure also lowers the Theorem 1 upper bound by replacing its dependence on the total number of environments $k$ with a dependence on $k_{\max}$ (the size of the largest cone).

**Theorem 2** (Regret bound under latent-space partitioning). *Let $h_1, \ldots, h_L \in \mathbb{S}^{d-1}$ be the context directions from $f_{dyn}$ and let $\{\mathcal{C}_j\}_{j=1}^L$ be disjoint cones around them. Assume block-separable parameterization (Assumption 2 in Appendix B), so that losses from $z \in \mathcal{C}_j$ depend only on block $(F_j, B_j)$. If $k_{\max} = \max_j |\{i : z_i \in \mathcal{C}_j\}|$, then*

$$\varepsilon_k^* \;=\; \max_{1 \leq j \leq L} \; \varepsilon_{|\mathcal{C}_j|}^* \;\leq\; \varepsilon_{k_{\max}}^*,$$

*and Theorem 3 holds with $\varepsilon_k^*$ replaced by $\max_j \varepsilon_{|\mathcal{C}_j|}^*$. (See Appendix B, Theorem 4.)*

Intuitively, Theorem 2 states that after the partitioning procedure of the latent space into non-overlapping clusters based on context representations $h$, the global worst-case FB approximation error $\epsilon_k = \max_{j \leq L} \epsilon_j$ is determined only by the cluster whose error $\epsilon_j$ is largest. Importantly, the bound depends on $k_{\max}$ rather than the total $k$. When $k_{\max}$ is controlled (e.g., via non-overlapping cones induced by an appropriate concentration $\kappa$), the bound becomes effectively independent of $k$. Full proof can be found in Appendix B

> **Takeaway 3**
>
> Adjusting the prior over task vectors $z_{\text{FB}}$ further mitigates the averaging effect and disentangles policy representations better based on the inferred dynamics.

## 4 EXPERIMENTS

In this section, we compare proposed methods, namely: **Belief-FB (BFB)** (Section 3.2) and its extension **Rotation-FB (RFB)** (Section 3.3), against the baselines in discrete and continuous settings. We outline experiments design below; all other necessary details are provided in Appendix D. Every environment is framed as a contextual MDP (CMDP), where the context differs by the underlying hidden variation (*e.g.*, grid layout, transition dynamics). During test time, we provide a single trajectory from random exploration policy, which enables context inference.

### 4.1 ENVIRONMENTS AND SETUP

To support claims and theoretical insights made in previous sections, we consider the following experimental setups: **(i)** discrete, partially observable Randomized Four-Rooms (Appendix C.2), **(ii)** continuous AntWind (Appendix C.3), and lastly **(iii)** continuous partially observable Randomized-Pointmass (Appendix C.4). We vary the number of train layouts for each experiment, while fixing the number of held-out *unseen* context settings to 20 for Randomized Four-Rooms and Randomized-Pointmass, and 4 for Ant-Wind. We perform comparisons against following baselines:

**HILP** (Park et al., 2024) is a method that learns state representations from offline data so that the distance in the learned representation space is proportional to the number of steps between two states in original space. **FB** (Touati & Ollivier, 2021) is an original version of the FB, described in Section 2. **Laplacian RL (LAP)** (Wu et al., 2019) constructs a graph Laplacian over state transitions from experience replay, then computes its eigenvectors to form low-dimensional representations that capture the environment's intrinsic structure. **Random** agent, which randomly explores the environment in a task-independent manner.

**Randomized Four-Rooms** is a discrete, deterministic, partially observable environment, where the task is to optimally move to the goal location. Training data is collected by executing random policies in $N$ distinct grid layouts, that differ in doorway and wall locations.

**Ant-Wind** is a continuous environment, where the goal is to make an ant to walk forward as fast as possible. The environment dynamics are determined by the direction (angle) of a wind $d$.

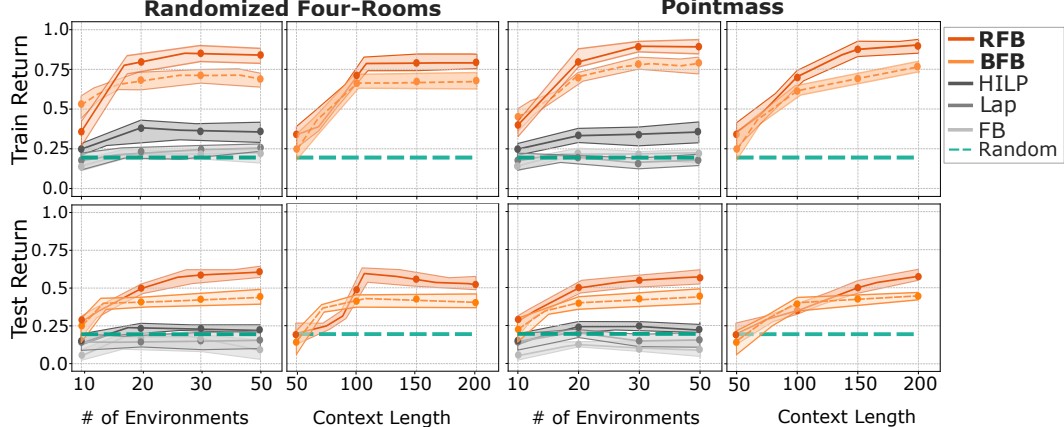

Figure 5: **Ablations on data diversity and context length of transformer encoder.** We show the influence of number of environments (data diversity) and context length on train and test performance in Four-Rooms and Pointmass environments. For data-diversity ablation, we see a clear performance boost up until some point, after which it plateos, as the Theorem 1 predicts. In our context-length ablation, we observe similar behaviour: performance improves as the context grows up to the length of a single episode, and then levels off. The results are averaged across three seeds, the opaque fill indicates standard deviation.

**Randomized-Pointmass** is a partially observable continuous environment, where the task is to move to the goal locations. Maze grid structure is generated randomly, where each cell either contains wall or empty, while ensuring there is a path between start and goal locations.

### 4.2 COULD

BELIEF ESTIMATION ENABLE ADAPTATION IN FB?

Previously, we provided the theoretical foundations and speculated on the matter why FB is unable to differentiate between distinct dynamics and how we can use the belief estimation to overcome this. We refer to Table 2 and Figure 1 that show our empirical findings to support our claims.

We would like to point out that neither FB nor LAP are able to outperform a simple random baseline in PointMass and Four-Room, indicating that the policy they learn is most likely stuck in some obstacle due to averaging (see Section 3.1. Only HILP, which uses a different way to learn policy representations, is able to perform better than random policy.

Belief-FB and Rotation-FB outperform every baseline method, indicating that belief estimation is indeed a missing piece for adaptation. Notably, our methods also demonstrate generalization capabilities beyond train data on unseen test tasks.

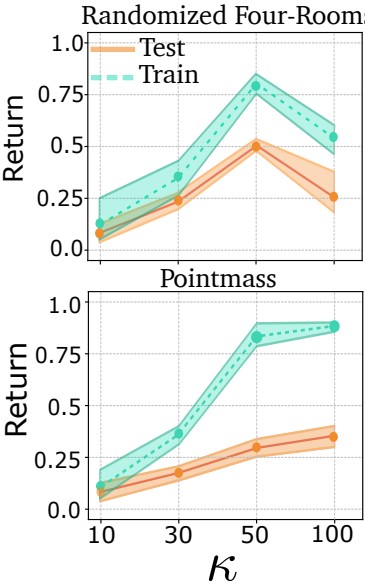

Figure 6: **Influence of $\kappa$ in RFB on performance.** The results are averaged across three seed, the opaque fill represents standard deviation.

### 4.3 DO BFB AND RFB CAPTURE HIDDEN PROPERTIES OF THE ENVIRONMENT?

For an agent to refine its policy, it needs to keep track and update the uncertainty over possible environment configurations. Both Belief-FB and Rotation-FB accomplish this. Figure 7 illustrates this phenomenon visually. In Randomized-Door (left), the episodic trajectories from five layouts form non-overlapping clusters in the first two principal components of $h$, effectively disentangling different dynamics.

In Ant-Wind, the embeddings lie almost perfectly on a circle whose azimuth matches the underlying wind direction, generalizing smoothly to the 4 held-out wind angles. The quantitative results for evaluation in Table 2 (averaged across all environments) reveal that the baseline methods fail to

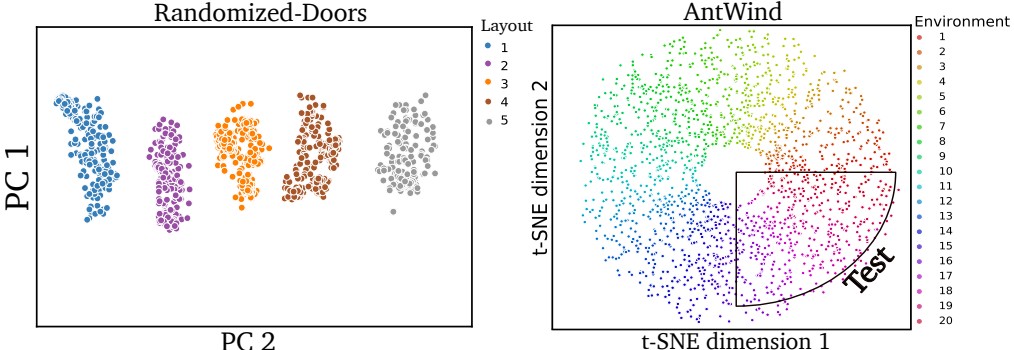

Figure 7: **2D projections of $z_{\text{dyn}}$ inferred from different trajectories across number of different contexts (colors), showing effective disentangling environments based on transition function or other mismatches.** (*left*) First two principal components are visualized for estimated $z_{\text{dyn}}$ from five trajectories, each representing different layout type in Randomized-Doors. (*right*) Inferred context variables $z_{\text{dyn}}$ recover hidden wind direction parameter in AntWind environment both for train and test, proving successful extrapolation properties.

recover those environment-specific properties and therefore produce sub-optimal policies even for train cases. In particular, HILP tends to predict an average direction in Randomized Four-rooms and ignores obstacles, while FB outputs same policy and $Q$ function for almost all environments. Figure 12 shows that $Q$ function is properly estimated only for BFB and RFB, respecting wall positions.

### 4.4 DOES CHANGE IN CONTEXT LENGTH INPUT TO THE $f_{\text{DYN}}$ IMPACTS PERFORMANCE?

In this experiment, we examine whether increasing the input trajectory length of improves performance. We vary the context length of $f_{\text{dyn}}$ from 50 to 200 and present the results in Figure 5 for both Randomized Four-Rooms and Randomized Pointmass environments, across train and test configurations. The results show that performance is poor when the context length is shorter than a single trajectory episode (100 steps), as short trajectories only capture local, near-term goals. Conversely, excessively long sequences provide no additional benefit due to redundancy, since $f_{\text{dyn}}$ already contains all neccessary information. Evaluations on both train and test environments demonstrate that $f_{\text{dyn}}$ produces representations $h$ capable of distinguishing between different context instances while maintaining robustness.

### 4.5 DOES INCREASE IN DATASET DIVERSITY MAKE POLICIES MORE ROBUST?

We investigate if diversifying CMDP training configurations improves performance. Intuitively, broader state-action space coverage enhances successor measure estimation. Experiments confirm this: Figure 5 shows rapid improvement for BFB up to 25 configurations, while baselines match random policy performance. Once learned representations $h$ from $f_{\text{dyn}}$ cover all variation modes (contexts), additional data yields minimal gain ($< 3\%$). These results align with Theorem 1.

### 4.6 HOW $\kappa$ IN RFB INFLUENCES PERFORMANCE?

As described in Section 3.3, RFB concentration $\kappa$ regularizes the diversity of policies for each environment. One the one hand, concentration should be high to ensure non-overlapping policy parametrized clusters $\pi_z$ for different $h$, while at the same time it should not exceed certain value to control the diversity of policies in the environment, preventing collapsed solutions. Figure 6 shows that lower values of $\kappa$, meaning task-vectors $z_{\text{FB}}$ are sampled with high deviation around $h$, likely producing overlapping clusters. As $\kappa$ grows, task-vectors become more specialized, lowering variance which results in higher performance.

## 5 CONCLUSION & LIMITATIONS

In this paper, we introduce **Belief-FB (BFB)** and **Rotation-FB (RFB)**, two methods that extend the Forward-Backward representations to handle dynamics mismatches, bridging the gap towards truly adaptive agents. At first, we identify a critical limitation in existing BFMs both theoretically

Table 1: Zero-shot performance across environments with varying dynamics. Results for FourRooms, PointMass, and AntWind are aligned with the main paper. We add Oracle-ID (one-hot environment ID concatenation) and Contextual-FB (our reimplementation of Jeen & Cullen (2024)). Oracle-ID excels in-distribution but fails to generalize out-of-distribution (OOD). Contextual-FB underperforms due to reliance on classifier expressivity. For the new OGBench Scene environment, we vary friction from 0.4-1.0 (train) and test on unseen low friction 0.1-0.3, demonstrating dynamics generalization akin to AntWind (wind direction variation). Higher is better.

| Method | FourRooms | | PointMass | | AntWind | | OGBench Scene | |
|---|---|---|---|---|---|---|---|---|
| | Train | Test | Train | Test | Train | Test | Train | Test |
| FB | $0.25 \pm 0.05$ | $0.15 \pm 0.04$ | $0.20 \pm 0.05$ | $0.10 \pm 0.03$ | $390 \pm 40$ | $250 \pm 30$ | $0.40 \pm 0.06$ | $0.20 \pm 0.05$ |
| LAP | $0.20 \pm 0.04$ | $0.10 \pm 0.03$ | $0.15 \pm 0.04$ | $0.10 \pm 0.03$ | $340 \pm 35$ | $290 \pm 25$ | $0.30 \pm 0.05$ | $0.10 \pm 0.03$ |
| HILP | $0.40 \pm 0.06$ | $0.20 \pm 0.05$ | $0.45 \pm 0.06$ | $0.25 \pm 0.05$ | $410 \pm 45$ | $410 \pm 40$ | $0.50 \pm 0.07$ | $0.30 \pm 0.06$ |
| Contextual-FB | $0.35 \pm 0.05$ | $0.18 \pm 0.04$ | $0.30 \pm 0.05$ | $0.15 \pm 0.04$ | $450 \pm 50$ | $350 \pm 40$ | $0.60 \pm 0.08$ | $0.40 \pm 0.07$ |
| Oracle-ID | $0.90 \pm 0.03$ | $0.10 \pm 0.03$ | $0.92 \pm 0.02$ | $0.08 \pm 0.02$ | $780 \pm 30$ | $50 \pm 20$ | $0.95 \pm 0.02$ | $0.0 \pm 0.02$ |
| BFB (ours) | $\mathbf{0.70 \pm 0.07}$ | $0.40 \pm 0.06$ | $0.76 \pm 0.07$ | $0.45 \pm 0.06$ | $680 \pm 60$ | $550 \pm 50$ | $0.6 \pm 0.07$ | $0.45 \pm 0.06$ |
| RFB (ours) | $\mathbf{0.85 \pm 0.04}$ | $\mathbf{0.61 \pm 0.05}$ | $\mathbf{0.88 \pm 0.04}$ | $\mathbf{0.55 \pm 0.05}$ | $\mathbf{740 \pm 40}$ | $\mathbf{640 \pm 40}$ | $\mathbf{0.7 \pm 0.04}$ | $\mathbf{0.55 \pm 0.05}$ |

and empirically: interference arises when training procedure relies on naively sampling policy-encoding latent directions for transition samples coming from conflicting dynamics. To address this, we learn hidden context variables (belief states) via a transformer encoder and use them as additional conditioning (Belief-FB) to the FB. Further, we improve latent-direction sampling by aligning task-relevant abstractions with environment-specific context, ensuring non-overlapping distinct environment specific regions in policy-encoding latent space. Both BFB and RFB demonstrate theoretical and empirical improvements over prior methods. However, limitations include evaluations on a narrow set of dynamics mismatches and the introduction of the additional hyperparameters, such as $\kappa$ that controls policy diversity across environments. Moreover, both BFB and RFB inherit all of the drawbacks of the vanilla FB representations: limitation to only linear reward representations and convergence guarantees. Also, random exploration at test time could fail at more complex environments and combining BFB and RFB together with more clever exploration methods at test time (Grillotti et al., 2024; Urpí et al., 2025) would make methods more scalable.

As future research directions, it would be valuable to investigate whether other zero-shot RL methods, those not based on successor-measure estimation, exhibit similar interference issues, and to scale our approach to more complex benchmarks such as XLand-MiniGrid (Nikulin et al., 2024; 2025) or Kinetix (Matthews et al., 2025).

## 6 ACKNOWLEDGMENTS

This work was supported by the The Ministry of Economic Development of the Russian Federation in accordance with the subsidy agreement (agreement identifier 000000C313925P4H0002; grant No 139-15-2025-012).

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

## A    EXTENDED RELATED WORKS AND BACKGROUND

### A.1    BACKGROUND

**Contexual Markov Decision Process.** Throughout paper we will be dealing with a Contextual Markov Decision Process (CMDP), defined by a tuple $\langle \mathcal{C}, \mathcal{S}, \mathcal{A}, \gamma, \mathcal{M} \rangle$, where $\mathcal{C}$ is a context space and $\mathcal{S}, \mathcal{A}$ are shared state and action spaces across environments. Function $\mathcal{M}$ maps particular context $c \in \mathcal{C}$ to respective MDP, *i.e.*, $\mathcal{M}(c) = \langle \mathcal{S}, \mathcal{A}, \mathcal{T}^c, R^c, \mu^c, \gamma \rangle$ with context-dependent transition function $\mathcal{T}^c : \mathcal{S} \times \mathcal{A} \times \mathcal{C} \to \mathcal{S}$, $\mu^c$ being an initial distribution over states and $\gamma \in (0, 1)$ a discount factor. Intuitively, the context $c \in \mathcal{C}$ represents a fixed environmental configuration, such as obstacle positions, layout geometry, dynamics vector parameters or seed. Throughout this work, the context remains static within each episode, consistent with prior literature (Modi et al., 2018; Kirk et al., 2023; Teoh et al., 2025). A policy $\pi : \mathcal{S} \to \Delta\mathcal{A}$ is optimal for context $c$ for the reward function $R$ if it maximizes expected discounted future reward, *i.e.*, $\pi^*_{c,R}(s_0, a_0) = \arg\max_\pi \mathbb{E}[\sum \gamma^t R(s_t, a_t)|s_0, a_0, \pi, c]$.

When the context is fully observable, augmenting the state space with the given context reduces the CMDP to a standard MDP, eliminating the need to model distinct dynamics $\mathcal{T}^c$, rewards $R^c$ or initial states $\mu^c$. However, if the context is partially observable, the learned model must infer and track the uncertainty over true hidden configuration to maintain theoretical optimality guarantees. Such task can be framed as posterior estimation $p(c|\mathcal{H})$ or *belief* over possible contexts $c$ given accumulated history $H$.

Most successful methods for deriving an optimal policy across arbitrary tasks from a task-agnostic dataset leverage successor features (Dayan, 1993; Barreto et al., 2017; Borsa et al., 2019; Park et al., 2024; Zhu et al., 2024) or their continuous counterpart, successor measures (Blier et al., 2021; Touati & Ollivier, 2021; Touati et al., 2022; Agarwal et al., 2025; Jeen et al., 2024). In this work, we focus on the latter framework, specifically its instantiation via forward-backward representations (Touati & Ollivier, 2021). Below, we briefly outline its key properties.

**Zero-Shot RL.** Given an offline dataset of transitions $\mathcal{D} = \{(s_i, a_i, s_{i+1})\}_{i=1}^{|\mathcal{D}|}$ generated by an unknown behavior policies, the agent's objective is to learn a compact abstraction of the environment from which it is possible . At test time, this abstraction helps to obtain optimal policy for *any* reward function $r_{test}$ which defines a particular *task*. Reward function can be specified either as a small dataset of reward-labeled states $\mathcal{D}_{test} = \{(s_i, r_{test}(s_i))\}_{i=1}^k$ or as a direct mapping $s \to r_{test}(s)$. While some prior works assume access to the context labels (Gregor et al., 2019), we focus on the setting where the context is unknown and must be inferred from the data. Alternative formulations of zero-shot RL exist under other formalisms, and we refer to (Kirk et al., 2023) for comprehensive overview.

### A.2    RELATED LITERATURE

**Domain Adaptation and Transfer Learning in RL.** While our work will focus on domain adaptation applied to estimating successor measure for various dynamics mismatches, we start by briefly reviewing more general ideas in classic domain adaptation and refer to (Kouw & Loog, 2019) for detailed overview. Most methods for domain adaptation can be categorized into *importance-weighting* (Bickel et al., 2007; Uehara et al., 2016; Sønderby et al., 2016) and *domain-invariant feature learning* (Fernando et al., 2013; Eysenbach et al., 2021; Xing et al., 2021; Zhang et al., 2020) approaches. Former methods estimate the likelihood ratio of examples under samples from target domain versus samples from source, which is then used to recalibrate examples from the source domain. The latter approaches learn a unified representation of the environment, targeting to extract only task-relevant abstraction, negating distracting information.

The most relevant approach which enables FB representations to generalize across dynamics is *Contexual FB* (Jeen & Cullen, 2024). This approach uses importance-weighting formalism and introduces two classifiers, which estimate the likelihood of transitions $(s_t, a_t)$ and $(s_t, a_t, s_{t+1})$ being from train or test context and augment the reward function to account for those discrepancies in the dynamics. If augmented reward function lies in the linear span of the $\mathcal{Z}$ space during FB training, then the policy can be extracted as described in Equation 3. However, such an approach requires training classifiers from scratch for each novel layout of the environment, limiting its applicability.

**Meta-RL.** Another major line of related works, Meta-Reinforcement Learning (Meta-RL), focuses on few-shot domain adaptation to unseen tasks or dynamics (Beck et al., 2024). The significant part of research in Meta-RL is dedicated to explicitly learning the *belief* by collecting a history of interactions with the environment on inference during test-time (Zintgraf et al., 2020; Dorfman et al., 2021; Rakelly et al., 2019). However, recent works show that it is possible to quantify the *belief* without learning the posterior implicitly (Laskin et al., 2022; Lee et al., 2023; Zisman et al., 2024; Sinii et al., 2024; Zisman et al., 2025; Tarasov et al., 2025; Polubarov et al., 2025). Leveraging in-context ability of transformers Vaswani et al. (2017), one can learn an end-to-end supervised model, while the transformer's context will absorb into robust representation the adaptation-relevant information thus enabling fast adaptation. We also leverage this in-context ability to construct the belief representation of the dynamics the agent currently in, but instead operating in a zero-shot manner.

## B PROOFS

=

**Notation recap.** Let $M^\pi(s, a, \cdot)$ be the successor measure of policy $\pi$ and $\rho$ the reference state–action measure used by FB training. As in the main text, FB seeks low-rank factors $F, B$ such that

$$M^\pi(s, a, \mathrm{d}s'\mathrm{d}a') \approx F(s, a, z)^\top B(s', a')\, \rho(\mathrm{d}s'\mathrm{d}a')$$

for policies $\pi = \pi_z$. For a set of $k$ CMDPs with optimal policies $\{\pi_i^\star\}_{i=1}^k$ and successor measures $\{M^{\pi_i^\star}\}_{i=1}^k$ we define the *worst-case class approximation error*

$$\varepsilon_k^\star := \inf_{F,B} \max_{1 \leq i \leq k} \left\| M^{\pi_i^\star} - F(\cdot, \cdot, z_i)^\top B(\cdot) \right\|_{L^2(\rho)}.$$

We write $\widehat{F}, \widehat{B}$ for the trained factors and set the (finite-sample / optimization) training discrepancy

$$\Delta_{\mathrm{est}} := \max_{1 \leq i \leq k} \left\| \widehat{F}(\cdot, \cdot, z_i)^\top \widehat{B}(\cdot) - F^\star(\cdot, \cdot, z_i)^\top B^\star(\cdot) \right\|_{L^2(\rho)},$$

where $(F^\star, B^\star)$ is a minimizer in the definition of $\varepsilon_k^\star$ (any minimizer will do). Unless otherwise noted we evaluate expectations w.r.t. a test distribution $\rho_{\mathrm{test}}$ that is absolutely continuous w.r.t. $\rho$ (Assumption 1 in the main paper), with density ratio bounded by $\kappa := \sup_{s,a} \frac{\mathrm{d}\rho_{\mathrm{test}}}{\mathrm{d}\rho}(s, a) < \infty$.

**Lemma 1** (Uniform successor-to-value stability). *Suppose that for some $\varepsilon \geq 0$,*

$$\sup_{(s_0, a_0)} \left\| F(s_0, a_0, z_R)^\top B(\cdot) - \frac{M^{\pi_{z_R}}(s_0, a_0, \cdot)}{\rho(\cdot)} \right\|_{L^2(\rho)} \leq \varepsilon.$$

*Then for any bounded reward $\|r\|_\infty \leq R$, $\| Q_r^* - Q_r^{\pi_{z_R}} \|_\infty \leq \frac{3}{1-\gamma} R\varepsilon.$*

*Proof sketch.* By standard successor-occupancy identities, $Q_r^\pi(s_0, a_0) = \int r(s, a)\, M^\pi(s_0, a_0, \mathrm{d}s\mathrm{d}a)$. The linear functional $M \mapsto \int r\, \mathrm{d}M$ has operator norm $\leq \|r\|_\infty$. Combining the uniform $L^2(\rho)$ error on $M/\rho$ with the contraction of the Bellman resolvent yields the stated $(3/(1-\gamma))R$ factor (details as in the cited stability proofs; constants unchanged). □

**Theorem 3** (Regret bound for multiple dynamics with decoupled errors). *Under Assumption 1 and for any bounded reward $\|r\|_\infty \leq R$, the policy extracted from the trained factors for CMDP $i$ (namely $\pi_{z_i}$ with $z_i$ computed from $r$ and $\widehat{B}$) satisfies*

$$\mathbb{E}_{(s,a) \sim \rho_{\mathrm{test}}}\left[ Q_r^*(s, a) - Q_r^{\pi_{z_i}}(s, a) \right] \leq \frac{3}{1-\gamma} R\left(\varepsilon_k^* + \Delta_{\mathrm{est}}\right).$$

*Moreover, $\varepsilon_{k+1}^* \geq \varepsilon_k^*$ (monotonicity in $k$).*

*Proof.* Applying Lemma 1 with $\varepsilon = \varepsilon_k^* + \Delta_{\mathrm{est}}$ yields $\|Q_r^* - Q_r^{\pi_{z_i}}\|_\infty \leq \frac{3}{1-\gamma} R(\varepsilon_k^* + \Delta_{\mathrm{est}})$. Taking expectation gives the displayed inequality since $\mathbb{E}_{\rho_{\mathrm{test}}}[f] \leq \|f\|_\infty$. Monotonicity is immediate because $\max$ over a larger index set cannot decrease. □

**Discussion (Theorem 1).** The upper bound separates an *intrinsic* model-class term $\varepsilon_k^\star$ (harder when more heterogeneous CMDPs are included) from a *finite-sample/optimization* term $\Delta_{\text{est}}$ (which can shrink with more data). Thus, adding CMDPs enlarges the worst-case *approximation class* but may still reduce empirical regret if $\Delta_{\text{est}}$ decreases.

**Assumption 2** (Block-separable parameterization). There exists a partition $\{\mathcal{S}_j\}_{j=1}^L$ of task directions and a routing function $g : \mathcal{Z} \to [L]$ such that the model uses disjoint parameter blocks $(F_j, B_j)$: for $z \in \mathcal{S}_j$ the prediction is $F_j(s, a, z)^\top B_j(\cdot)$ and no other block parameters are used.

**Theorem 4** (Decoupling under block-separable parameters). *Assume Assumption 2. Let $k_{\max} = \max_j |\mathcal{S}_j|$. Then the training objective decouples across blocks $j$, and the worst-case uniform class error satisfies*

$$\varepsilon_k^* \;=\; \max_{1 \le j \le L} \; \varepsilon_{|\mathcal{S}_j|}^* \;\le\; \varepsilon_{k_{\max}}^*.$$

*Consequently, the regret bound in Theorem 1 depends on $k_{\max}$ (not on $k$).*

*Proof (sketch).* By Assumption 2, losses from tasks $z \in \mathcal{S}_j$ depend only on $(F_j, B_j)$, hence the empirical and population objectives decompose as a sum $\sum_{j=1}^L \mathcal{L}_j(F_j, B_j)$. Minimizers are obtained by solving each block independently. The definition of $\varepsilon_m^*$ as the optimal uniform $L^2(\rho)$ error over $m$ tasks then yields $\varepsilon_k^* = \max_j \varepsilon_{|\mathcal{S}_j|}^* \le \varepsilon_{k_{\max}}^*$. $\qquad\square$

**Discussion (Theorem 2).** Partitioning $z$ into disjoint cones removes interference: optimization decouples by block, so adding new cones does not inflate the worst-case error beyond the hardest block. Practically, once $F, B$ have enough capacity for the largest block ($d \ge k_{\max}$ in a tabular analogy), the class error can be driven to zero *without* growing with $k$.

Let $\{M_{\pi_i}\}$ be a collection of successor measure of the optimal policies $\{\pi_i\}_{i=1}^k$ for $k$ distinct CMDPs. Given a reference measure $\rho$ on $S \times A$, define the worst-case *class approximation error* as

$$\epsilon_k := \inf_{F,B} \max_{i \le i \le k} ||M_{\pi_i} - F(\cdot, \cdot, z_i)^T B(\cdot)||_{L_\rho^2} \tag{8}$$

### B.1 Forward–Backward (FB) training

**Successor measure and FB factorization.** For a policy $\pi$ and discount $\gamma \in (0, 1)$, the successor measure $M^\pi(s_0, a_0, \cdot)$ is the (discounted) future occupancy of next states,

$$M^\pi(s_0, a_0, X) \;=\; \sum_{t \ge 0} \gamma^t \Pr\big(s_{t+1} \in X \mid s_0, a_0, \pi\big), \qquad X \subseteq \mathcal{S},$$

and, for state-based rewards $r : \mathcal{S} \to \mathbb{R}$,

$$Q_r^\pi(s_0, a_0) \;=\; \int r(s^+) \, M^\pi(s_0, a_0, ds^+).$$

FB approximates $M^\pi$ (hence all $Q_r^\pi$) with a finite-rank factorization conditioned on a *task vector* $z \in \mathcal{Z} \subset \mathbb{S}^{d-1}$:

$$M^{\pi_z}(s, a, ds^+) \;\approx\; \big\langle F(s, a, z), \, B(s^+) \big\rangle \rho(ds^+),$$

where $F : \mathcal{S} \times \mathcal{A} \times \mathcal{Z} \to \mathbb{R}^d$ is the *forward* map, $B : \mathcal{S} \to \mathbb{R}^d$ the *backward* map, $\langle \cdot, \cdot \rangle$ denotes the Euclidean inner product, and $\rho$ is a reference distribution over next states drawn from the offline dataset. From the factorization it follows that

$$\begin{aligned} Q_r^{\pi_z}(s, a) \;&\approx\; \int r(s^+) \big\langle F(s, a, z), \, B(s^+) \big\rangle \rho(ds^+) \\ &=\; \big\langle F(s, a, z), \, z_r \big\rangle, \quad z_r \;\triangleq\; \mathbb{E}_{s^+ \sim \rho}\big[r(s^+) B(s^+)\big]. \end{aligned} \tag{9}$$

**Greedy policy family.** For each $z \in \mathbb{S}^{d-1}$, the *greedy* policy associated with the representation is

$$\pi_z(s) \;\in\; \arg\max_{a \in \mathcal{A}} \big\langle F(s, a, z), \, z \big\rangle. \tag{10}$$

In discrete action spaces we take the exact maximizer; in continuous control we use an actor network to approximate equation 10 (DDPG-style).

---

In some variants $B$ depends on $(s, a)$; our implementation uses $B(s)$ as in the original formulation.

**Bellman identity for the successor measure.** Let $s_{t+1} \sim T(\cdot \mid s_t, a_t)$ and $a_{t+1} \sim \pi_z(\cdot \mid s_{t+1})$. For any *anchor* $s^+ \sim \rho$, the successor measure satisfies

$$\underbrace{\frac{M^{\pi_z}(s_t, a_t, ds^+)}{\rho(ds^+)}}_{\text{"density" w.r.t. } \rho} = \mathbf{1}\{s^+ = s_{t+1}\} + \gamma \mathbb{E}\left[\frac{M^{\pi_z}(s_{t+1}, a_{t+1}, ds^+)}{\rho(ds^+)}\right]. \tag{11}$$

FB enforces this identity by regressing the scalar score $\langle F(\cdot), B(s^+)\rangle$ against the right-hand side across random anchors $s^+$.

**Training objective (anchor regression).** Given a dataset $D = \{(s_t, a_t, s_{t+1})\}$, sample $z \sim \mathcal{Z}$ (e.g., uniformly on $\mathbb{S}^{d-1}$ or from a mixture that also uses $B$), compute $a_{t+1} \approx \pi_z(s_{t+1})$ via equation 10, and draw anchors $s^+ \sim \rho$. Using target networks $\widehat{F}, \widehat{B}$ (Polyak-averaged), the FB loss is

$$\mathcal{L}_{\text{FB}} = \mathbb{E}_{(s_t, a_t, s_{t+1}) \sim D} \, \mathbb{E}_{z \sim \mathcal{Z}} \, \mathbb{E}_{s^+ \sim \rho} \left[ \left\langle F(s_t, a_t, z), B(s^+)\right\rangle - \mathbf{1}\{s^+ = s_{t+1}\} \right.$$

$$\left. - \gamma \left\langle \widehat{F}(s_{t+1}, a_{t+1}, z), \widehat{B}(s^+)\right\rangle \right]^2. \tag{12}$$

On a discrete replay buffer (finite $\rho$), expanding the square in equation 12 yields the practically convenient equivalent form

$$\boxed{\begin{aligned} \mathcal{L}_{\text{FB}} = \mathbb{E}_{(s_t, a_t, s_{t+1}, s^+) \sim D, \, z \sim \mathcal{Z}} \Big[ &\left(\langle F(s_t, a_t, z), B(s^+)\rangle - \gamma\langle\widehat{F}(s_{t+1}, a_{t+1}, z), \widehat{B}(s^+)\rangle\right)^2 \\ & - 2\left\langle F(s_t, a_t, z), B(s_{t+1})\right\rangle \Big], \end{aligned}} \tag{13}$$

which we use in implementation. Gradients update $(F, B)$ while $\widehat{F}, \widehat{B}$ are updated by slow averaging. The actor (continuous actions) is trained to maximize $a \mapsto \langle F(s, a, z), z\rangle$.

**Zero-shot RL procedure (test-time).** FB is trained *without rewards* in an unsupervised regime. At test time, for a new task specified by a reward function $r$ (or a small set of labeled states $\{(s_i, r(s_i))\}$), we:

1. **Infer the task vector.** Form

$$z_r = \mathbb{E}_{s^+ \sim \rho}\left[r(s^+)\, B(s^+)\right]$$

2. **Act greedily w.r.t. $z_r$.** Use the policy $\pi_{z_r}$ in equation 10: $\pi_{z_r}(s) \in \arg\max_a \langle F(s, a, z_r), z_r\rangle$.

If $z_r$ lies (approximately) in the linear span of task vectors encountered during training, then $Q_r^{\pi_{z_r}}(s, a) \approx \langle F(s, a, z_r), z_r\rangle$ and $\pi_{z_r}$ is near-greedy for $Q_r$ in the sense of our analysis.

**Practical notes.** (i) We normalize $z, B(s)$ to the hypersphere for stability; (ii) we mix *uniform* and *backward-induced* sampling for $z$ during training; (iii) target networks and large anchor batches stabilize the regression in equation 12–equation 13; (iv) in continuous control we learn an actor (DDPG-style) to approximate the argmax in equation 10. The entire pipeline requires *no* reward labels during training, enabling zero-shot extraction for arbitrary test-time rewards.

## C  Environment Descriptions

### C.1  Randomized-Doors

The Randomized-Doors MiniGrid environment (Figure 8) is a discrete-state, discrete-action finite horizon deterministic environment in which agent has an objective to go to goal location with maximum return of 1. Each episode terminates after 100 steps or after reaching goal location. The randomization determines possible open doors locations, fully specifying particular layout. In our experiments, the observation state of an agent consists of $(x, y)$ coordinates tuple, making it partially observable. Such setting requires to properly update beliefs over unobservable layout configuration type. The action space consists of four actions, namely {up, down, right, left}, while $(x, y)$ coordinates across both axes are bounded by grid size, which we take to be $9 \times 9$.

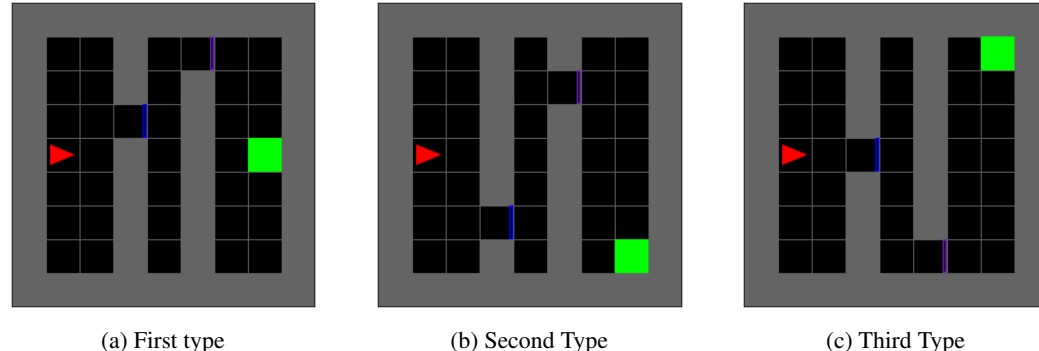

(a) First type                    (b) Second Type                    (c) Third Type

Figure 8: Several possible layouts are visualized, each corresponding to unique possible doors configurations. The agent is denoted as a red triangle. The task specification (goal position) with reward of 1 is denoted by green square and is also randomized. It is a custom implementation based on Empty MiniGrid (https://minigrid.farama.org).

## C.2 RANDOMIZED FOUR-ROOMS

The Randomized Four-Rooms MiniGrid environment Figure 9 is a modification of classic Four-Rooms and is a discrete-state, discrete-action, deterministic partially observable environment. For each episode, the maze layout (grid type) is generated randomly, ensuring all of the four rooms are connected with exactly single door. Observation state consists of $(x, y)$ coordinates, making this environment hard and checks whether agent could successfully estimate uncertainty over hidden configurations solely based on number of occurrence of each transition, recovering dynamics. In our experiments, we consider $11 \times 11$ bounds for height and width.

Observation space consists of raw discrete $(x, y)$ coordinates on the grid, while actions correspond to a set of possible moves {up, down, left, right}. For every layout we record 500 episodes of length 100, yielding a dataset that covers almost all possible $(s, a)$ transitions. For testing on unseen configurations, we fix agent starting position to coordinates of the first empty cell and evaluate performance across 3 static goal positions, farhest away from starting position.

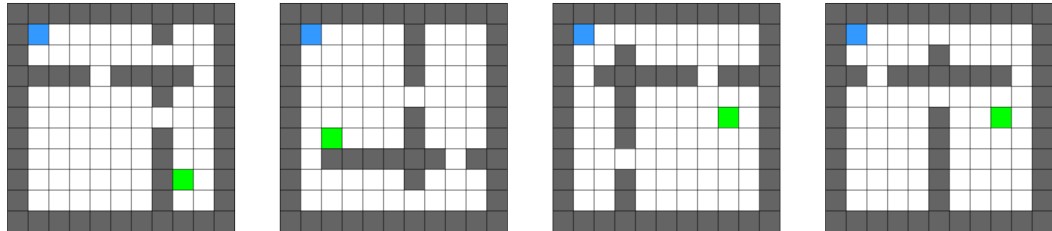

Figure 9: **Different layout configurations from randomized Four-Rooms environment.** During inference, the goal for the agent (depicted in blue) is to achieve green location. In our experiments we fix starting agent position and fix 3 goals, one for each room.

## C.3 ANT-WIND

The AntWind environment is a modified version of the Ant locomotion task from the MuJoCo simulator, commonly used to test an agent's adaptability to changing dynamics. In this environment, an ant-like robot must learn to move forward while being subjected to external wind forces varying in magnitude and direction. In our experiments we consider 17 environments for training, covering three quadrants of possible wind directions on the circle, while leaving others for test, checking extrapolation on the fourth quadrant.

For our experiment, we collect dataset by training SAC (Haarnoja et al., 2018) on 3/4 of all possible directions, which results in 16 environments and hold out the other 1/4 for evaluation. Resulting dataset consists of 3400 transition tuples, where each environment configuration is represented as trajectory of length 256.

## C.4 RANDOMIZED POINTMASS

Randomized Pointmass is a modification of pointmass environment from D4RL Fu et al. (2020). Each episode the environment grid structure is randomized, ensuring all cells are interconnected. The observation space consists of $(x, y)$ transitions. Start position is determined as a first empty cell, while goal location is chosen to be the fartherst away from start (based on Manhattan distance) and ensuring existence of at least one valid trajectory (*e.g.*, through BFS).

Observation space consists of $(\texttt{global } x, \texttt{global } y)$ position, similar to Four-Rooms. We fix dataset size to be $1e^6$, only varying number of layouts and episodes per layout, while fixing episode length to 250. We use explore policy, which is a random policy with a portion of actions repeated ("sticky-actions").

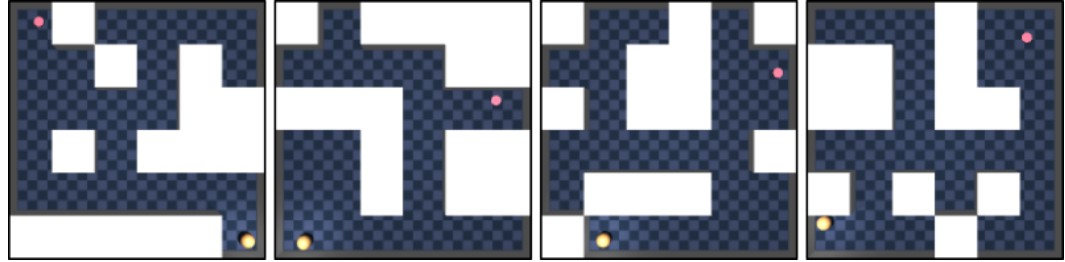

Figure 10: **Examples of pointmass grid variations.**

# D EXPERIMENTS DETAILS

**Randomized-Doors.** For didactic example from Section 3.1 we collect diverse dataset from different layout configurations (open door locations) such that visitation distribution over all states is non-zero. Black color denotes obstacles. The episode length is set to be 100, which is equal to the context length of the transformer encoder for this experiment. Overall, we collect 500 episodes per layout and coverage heatmap is visualized in Figure 11.

Table 2: Comparison of proposed approaches against baselines on **test** (unseen) environments. Results for Fourrooms and Pointmass are averaged across 20 mazes configurations.

| Environment (Test) | Method | | | | | |
|---|---|---|---|---|---|---|
| | Random | Vanilla-FB | HILP | Lap | **Belief-FB** | **Rotation-FB** |
| Randomized-Fourrooms | $0.05_{\pm 0.01}$ | $0.15_{\pm 0.06}$ | $0.2_{\pm 0.02}$ | $0.1_{\pm 0.1}$ | $0.4_{\pm 0.02}$ | $0.61_{\pm 0.02}$ |
| Randomized-Pointmass | $0.03_{\pm 0.01}$ | $0.1_{\pm 0.1}$ | $0.25_{\pm 0.02}$ | $0.1_{\pm 0.1}$ | $0.45_{\pm 0.05}$ | $0.55_{\pm 0.05}$ |
| Ant-Wind | $250_{\pm 200.0}$ | $250_{\pm 98.5}$ | $410_{\pm 40.5}$ | $290_{\pm 22.5}$ | $550_{\pm 50.5}$ | $640_{\pm 30.7}$ |

Table 3: Comparison of proposed approaches against baselines on **train** environments. Results for Fourrooms and Pointmass are averaged across 20 mazes configurations.

| Environment (Train) | Method | | | | | |
|---|---|---|---|---|---|---|
| | Random | Vanilla-FB | HILP | Lap | **Belief-FB** | **Rotation-FB** |
| Randomized-Fourrooms | $0.18_{\pm 0.02}$ | $0.25_{\pm 0.02}$ | $0.4_{\pm 0.02}$ | $0.2_{\pm 0.1}$ | $0.7_{\pm 0.02}$ | $0.85_{\pm 0.02}$ |
| Randomized-Pointmass | $0.0_{\pm 0.05}$ | $0.2_{\pm 0.2}$ | $0.45_{\pm 0.1}$ | $0.15_{\pm 0.15}$ | $0.76_{\pm 0.18}$ | $0.88_{\pm 0.2}$ |
| Ant-Wind | $-190_{\pm 250}$ | $390_{\pm 120}$ | $410_{\pm 90}$ | $340_{\pm 150}$ | $680_{\pm 80}$ | $740_{\pm 70}$ |

**Note on relience on random exploration during test time.** Random exploration relience of BFB and RFB in highly complex environments may fail to discover crucial states needed to disambiguate dynamics identification. However, we emphasize that our work addresses a distinct bottleneck: existing behavioral foundation models (BFMs), particularly FB, tend to collapse when trained on offline data composed of mixed CMDPs. Consequently, training BFMs on large scale mixed multi-modal (in terms of dynamics) data would yield an averaged policy, thus limiting their current

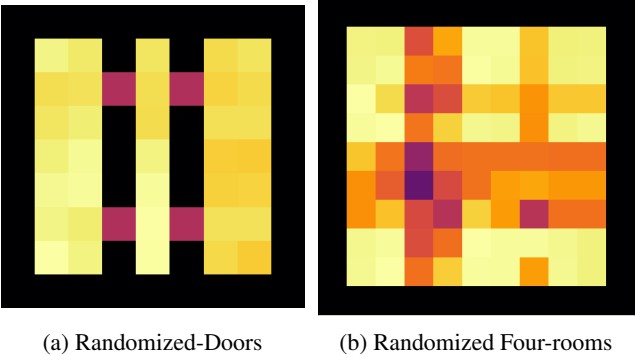

(a) Randomized-Doors        (b) Randomized Four-rooms

Figure 11: Empirical state occupancy measures ($\rho$) visualizations of collected datasets for discrete-based environments.

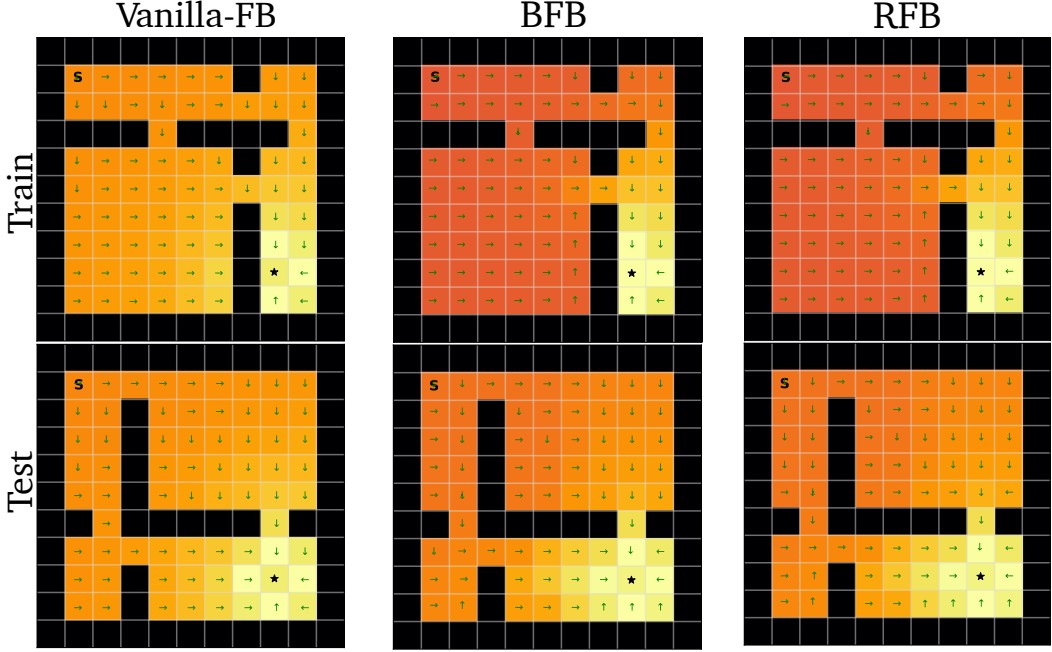

Figure 12: **Q-function and deterministic policy visualizations (Equation 3)** on Randomized Four-Rooms environment. Vanilla-FB ignores environment structure and resulting policy moves through obstacles. BFB and RFB do not have such issue.

applicability to unimodal datasets (in terms of dynamics mismatch). Both BFB and RFB overcome this collapse. Developing smarter test-time exploration strategies to streamline dynamics identification remains an important direction for future research.

## D.1    DATASET GENERATION

For Randomized Four-Rooms, we produce four training datasets with the following parameters:

| # Transitions | # layouts | # episodes per layout | episode length |
|---|---|---|---|
| 1000000 | 10 | 1000 | 100 |
| 1000000 | 20 | 500 | 100 |
| 1000000 | 30 | 250 | 100 |
| 1000000 | 50 | 150 | 100 |

Table 4: Details for Randomized Four-Rooms datasets

**Randomized Four-Rooms.** For experiments on Randomized Four-Rooms during dataset collection we generate randomly grid layout, ensuring that each room is interconnected by exactly one door. For evalution we fix agent start position to $(1, 1)$ with the goal of reaching 3 other goals, specified at other rooms. Each episode terminates after 100 steps. The evaluation protocol is averaged success rate across 3 across 20 environments.

**AntWind.** For AntWind we first collect trajectories by varying wind direction $d$ and training an expert-like SAC agent. After training, we collected evaluation trajectories from trained agent. This ensures that all directions are covered and explicitly sets dynamics context. As said in Experiments section, we train on 16 environments with wind directions corresponding to first 3 quadrants of circle, leaving other 4 (last quadrant) for hold out.

# E    IMPLEMENTATION DETAILS

## E.1    FORWARD-BACKWARD REPRESENTATIONS

### E.1.1    GPUS

We run each experiment on 1 Nvidia RTX 4090. The overall training time (for both dynamics encoder and FB training) is approximately 1 hour.

### E.1.2    ARCHITECTURE

The forward-backward architecture described below mostly follows the implementation by Touati et al. (2022). All other additional hyperparameters for BFB and RFB are reported in Table 5. Moreover, we should emphasize that our choice of transformer architecture for $f_{\text{dyn}}$ is mainly based on its abilities to encode large sequences, and other architectural designs (e.g State-Space Models, RNNs) can also be used. This choice does not change our observations from Section 3.2, Section 3.3.

**Forward Representation** $F(s, a, z)$. The input to the forward representation $F$ is always preprocessed. State-action pairs $(s, a)$ and state-task pairs $(s, z)$ have their own preprocessors which are feedforward MLPs that embed their inputs into a 512-dimensional space. These embeddings are concatenated and passed through a third feedforward MLP $F$ which outputs a $d$-dimensional embedding vector. Note: the forward representation $F$ is identical to $\psi$ used by USF so their implementations are identical (see Table 5). Also, for stability reasons of TD learning, we make ensemble of $F$ and take their `mean` as aggregation function.

**Backward Representation** $B(s)$. The backward representation $B$ is a feedforward MLP that takes a state as input and outputs a $d$-dimensional embedding vector.

**Actor** $\pi(s, z)$. Like the forward representation, the inputs to the policy network are similarly preprocessed. State-action pairs $(s, a)$ and state-task pairs $(s, z)$ have their own preprocessors which feedforward MLPs that embed their inputs into a 512-dimensional space. These embeddings are concatenated and passed through a third feedforward MLP which outputs a $a$-dimensional vector, where $a$ is the action-space dimensionality. A `Tanh` activation is used on the last layer to normalise their scale. Note the actors used by FB and USFs are identical (see Table 5). For discrete environments, optimal policy is greedy, while for continuous DDPG-style is used for approximating `argmax`.

**Misc.** Layer normalisation and `Tanh` activations are used in the first layer of all MLPs to standardise the inputs as recommended in original paper for both discrete and continuous becnhmarks. Baseline is taken from official repository `contrallable agent`.

Table 5: **Hyperparameters for FB.** Hyperparameters for Belief-FB and Rotation-FB are highlighted in

| Hyperparameter | Value |
|---|---|
| Latent dimension $d$ | 150 (100 for discrete) |
| $F$ / $\psi$ dimensions | (1024, 1024) |
| $B$ / $\varphi$ dimensions | (256, 256, 256) |
| Preprocessor dimensions | (1024, 1024) |
| Std. deviation for policy smoothing $\sigma$ | 0.2 |
| Truncation level for policy smoothing | 0.3 |
| Learning steps | 1,000,000 |
| Batch size | 1024 |
| Optimiser | Adam |
| Learning rate | 0.0001 |
| Learning rate of $f_{\mathrm{dyn}}$ | 0.0001 |
| Discount $\gamma$ | 0.99, 0.98 (Maze) |
| Activations (unless otherwise stated) | GeLU |
| Target network Polyak smoothing coefficient | 0.05 |
| $z$-inference labels | 10,000 |
| $z$ mixing ratio | 0.5 |
| $\kappa$ | 50, 100 for Pointmass |
| Contexual representation $h$ dimension | 150 (100 for discrete) |
| Next state predictor $g_{\mathrm{pred}}$ | (256, 256, 256) |

## E.2   HILP

We take official implementation in JAX from Park et al. (2024). All of the hyperparameters are the same as in original paper.

## E.3   TASK SAMPLING DISTRIBUTION $\mathcal{Z}$

**Vanilla-FB.** FB representations require a method for sampling the task vector $z$ at each learning step. Touati et al. (2022) employ a mix of two methods, which we replicate:

1. Uniform sampling of $z$ on the hypersphere surface of radius $\sqrt{d}$ around the origin of $\mathbb{R}^d$,

2. Biased sampling of $z$ by passing states $s \sim \mathcal{D}$ through the backward representation $z = B(s)$. This also yields vectors on the hypersphere surface due to the $L2$ normalization described above, but the distribution is non-uniform.

We sample $z \sim 50 : 50$ (either randomly or from $B$) from these methods at each learning step as in original work by Touati & Ollivier (2021).

**Rotation-FB.** After transformer $f_{\mathrm{dyn}}$ pretraining stage, RFB at each gradient step chooses task-conditioning vector $z_{\mathrm{FB}}$ based on **i)** context representation $h$ acting as axes coming from $f_{\mathrm{dyn}}$ and **ii)** drawing task encoding vectors $z_{\mathrm{FB}}$ around this axes. We also perform normalization as in Vanilla-FB by projecting resulting vector on a surface of hypersphere of radius $\sqrt{d}$.

**Stage ii)** is implemented as drawing samples as $z_{\mathrm{FB}} \sim \mathrm{vMF}(\mu = h, \kappa)$. In order to remove high computational costs, we implement this sampling procedure through Householder reflection around context axes, by first drawing $z$ from one of the basis vectors (*e.g.*, north pole) and then performing rotation.

### E.4 PSEUDOCODE

---
**Algorithm 1** Belief-FB Training

---
1: **Input**: offline diverse dataset $\mathcal{D}$ consisting of transitions based on hidden configuration variable $c_i$
2: Initialize transformer encoder $f_{\mathrm{dyn}_\theta}$, $F_\eta$, $B_\omega$, number of gradient steps for transformer pre-training $K$, context length $T$, Polyak coefficient, $\beta$, batch size $B$ learning rates $\lambda_f, \lambda_F, \lambda_B$
3: **while** update steps $< K$ **do**
4:     sample batch of $B$ trajectories of length $T$ $\{(s_{i,t}, a_{i,t}, s_{i,t+1})\}_{i=1,\ldots B, t=1,\ldots,T} \sim \mathcal{D}$
5:     $(\boldsymbol{\mu}_i; \log \boldsymbol{\sigma}_i), = f_{\mathrm{dyn}_\theta}(\{s_{i,t}, a_{i,t}, s_{i,t+1}\}_{t=1}^M), i = 1, \ldots, B,$
6:     $\boldsymbol{z_i} = \boldsymbol{\mu}_i + \boldsymbol{\epsilon}_i \odot \exp(\log \boldsymbol{\sigma}_i),$
7:     $\mathbf{Z}_{i,t} = \boldsymbol{z}_{\mathrm{dyn}_i}, \; t = 1, \ldots, T$    # Representation $z_{\mathrm{dyn}}$ is shared across each sequence
8:     $\hat{s}_{i,t+1} = g_{\mathrm{pred}}(s_{i,t}, a_{i,t}, \mathbf{Z}_{i,t}) \quad t = 1, \ldots, T, \; i = 1, \ldots, B$
9:     $\mathcal{L}_{\mathrm{context}} = \frac{1}{BT} \sum_{i=1}^{B} \sum_{t=1}^{T} \|\hat{s}_{i,t+1} - s_{i,t+1}\|_2^2$
10:     $\theta_{f_{\mathrm{dyn}}} \leftarrow \theta_{f_{\mathrm{dyn}}} - \lambda_f \nabla_\theta \mathcal{L}_{\mathrm{context}}(\theta)$
11: **end while**
12: **while** not converged **do**
13:     $\eta_F \leftarrow \eta_F - \lambda_F \nabla_{\eta_F} J_{(F,B)}(\eta_F)$    # FB training, Equation **??**
14:     $\omega_B \leftarrow \omega_B - \lambda_B \nabla_{\omega_B} J_{(F,B)}(\omega_B)$
15: **end while**

---

---
**Algorithm 2** Sampling $z_{\mathrm{FB}}$ for RFB

---
**Input:** $B$ (batch size), $d$ (latent dimension), anchor matrix $\mathbf{H} \in \mathbb{R}^{B \times d}$, $\kappa$ (concentration)
**Output:** $\mathbf{Z} \in \mathbb{R}^{B \times d}$
1: **Normalize anchors:** $\mathbf{u}_i \leftarrow \mathbf{H}_i / (\|\mathbf{H}_i\|_2 + \varepsilon)$          $\triangleright$ for $i = 1, \ldots, B$
2: $\mathbf{S} \leftarrow \mathrm{VMF\_SAMPLE\_NORTHPOLE}(B, d, \kappa)$          $\triangleright$ draw $B$ VMF samples
3: **for** $i \leftarrow 1$ **to** $B$ **do**
4:     $\mathbf{R}_i \leftarrow \mathrm{HOUSEHOLDER\_ROTATION}(\mathbf{u}_i)$
5:     $\mathbf{z}_i \leftarrow \mathbf{R}_i \mathbf{S}_i$
6: **end for**
7: $\mathbf{Z} \leftarrow \mathrm{PROJECT\_TO\_SPHERE}(\{\mathbf{z}_i\}_{i=1}^B)$
8: **return** $\mathbf{Z}$

---

