# OpenReview forum: "Zero-Shot Adaptation of Behavioral Foundation Models to Unseen Dynamics"
_ICLR.cc/2026/Conference — ICLR 2026 Poster_

### Official Review · Reviewer_dycu · 2025-10-20

**Soundness:** 3
**Presentation:** 3
**Contribution:** 3
**Rating:** 8
**Confidence:** 4

**Summary:**

The paper introduces an algorithm for unsupervised zero-shot reinforcement learning under partial observability. It considers environments where the dynamics are governed by a latent parameter that is not directly observable but can be inferred from transitions. The authors propose using a permutation-invariant, transformer-based architecture to build an estimate (or belief state) of the dynamics from a set of transitions. This belief vector is then used to condition the Forward-Backward model, i.e., both the actor and critic are trained conditioned on the belief state. Furthermore, they propose modifying the structure of the task/policy space to be belief-state dependent, resulting in clusters of dynamics. The effectiveness of the method is evaluated on three toy domains.

**Strengths:**

- As far as I know, this is the first paper to tackle the problem of unsupervised reinforcement learning in partially observable contextual Markov Decision Processes (MDPs).
- The approach is simple and sound, and the authors provide a clear, step-by-step construction of the method, explaining why each component is necessary.
- Although the core idea is not very novel, I believe its application to reinforcement learning is new and interesting for the community.

**Weaknesses:**

I don't see any major weaknesses to mention in this section. I will report a few questions in the next section.

**Questions:**

- I found Section 3.1 a bit trivial. It is expected that a method designed for a single MDP does not perform well in the considered contextual MDP. What am I missing? Why spend so much time justifying this aspect?
- I like the approach of building the latent belief by predicting the next state. Could you discuss further the choice of using a non-causal transformer? Why did you use Gaussian regularization?
- Do you use samples from the training dataset to perform reward inference? Does the performance change if you use transitions from the test environment for inference?
- You assume access to a set of transitions from the test environment to infer the context. However, you could also compute the context directly while interacting with the environment. Could you evaluate the test performance with respect to the number of samples used for belief inference?

---

> ### Author Response · Authors · 2025-11-18
>
> **Q1: Section 3.1 clarifications**
>
> **Forward-Backward representations, in theory, can capture any policies in the environment, which give non-zero occupancy measure for some states. So some latent directions can be assigned to distinct policies, which induces occupancy measure for various dynamics.** However, as we show in paper, theoretically and empirically why exactly this is not the case and what are the underlying issues.
>
> Section 3.1 formalizes and quantifies presence of interference. We show that when FB is trained on mixtures of dynamics without contextual conditioning, the learned successor measure averages over futures, entangling policy directions and yielding ambiguous greedy policies, even on training contexts (Figures 2-3). This is captured by Theorem 1, where the regret upper bound monotonically loosens with the number of environments $k$ via the worst-case class error $\epsilon^\star_k$ (monotone in $k$). the phenomenon is therefore structural, not an artifact of implementation. This analysis motivates Sections 3.2–3.3, where belief-conditioning and cone-wise sampling remove the interference and replace the explicit $k$ dependence by $k_{\text{max}}$ (Theorem 2).
>
> **Q2: Why a non-causal transformer for belief? Why Gaussian regularization?**
>
> We discard ordering and treat the history as a set of transitions because the hidden context is fixed within an episode; the encoder should be permutation-invariant and focus on dynamics, not policy idiosyncrasies. A non-causal permutation-invariant transformer can pool information from all positions to infer a single episode-level context vector $h$, which we then concatenate with $z$ for FB.
>
> We place a Gaussian prior/regularizer on $h$ to stabilize inference and encourage a smooth, unimodal belief space that supports sampling and clustering. $h$ is shared across the episode and trained by predicting $s_{t+1}$ from $(s_t, a_t, h)$. Empirically, we keep $B$ shared (unconditioned) to avoid oversmoothing the value landscape. This design follows recent in-context adaptation results: end-to-end transformers can amortize belief inference from few transitions, which we leverage in a zero-shot setting.
>
> **Q3: Do you use training samples for reward inference? What if you use test transitions?**
>
> Reward inference in FB computes $z_{\text{test}} = \mathbb{E}[Br]$. In practice, we use states from the reference distribution $\rho$ (Replay buffer) with provided test rewards (same as in FB setting) as either a small labeled set or a reward function. Number of such labeled samples is a hyperparameter ($z$-inference labels).
>
> Using test-environment transitions to approximate the same expectation is not required by our theory. Under Assumption 1 (Coverage) and Lemma B.1, computing $z_\text{R}$ under $\rho$ yields the optimal policy in the exact case and a stability bound in the approximate case. Changing to $\rho_{\text{test}}$ only changes constants via $\kappa$ hyperparameter, which controls cluster spread of suitable policies.
>
> **Q4: You assume a short set of test transitions to infer context; what about computing context online? Can you show performance vs length of exploration trajectory?**
>
> Our evaluation already conditions on a short, reward-free test trajectory to infer $h$ and then acts zero-shot. This is equivalent to computing context during early interaction. We ablate context length and find that performance improves with more samples up to roughly one episode, then plateaus, matching the intuition that belief becomes identifiable with modest history. Moreover, we ablate the trajectory length hyperparameter, which can be found in Figure 5 (Context length).

---

### Official Review · Reviewer_mdha · 2025-10-31

**Soundness:** 3
**Presentation:** 4
**Contribution:** 2
**Rating:** 6
**Confidence:** 2

**Summary:**

This paper identifies a critical failure in Forward-Backward (FB) based Behavioral Foundation Models: training on mixed-dynamics data causes interference, as the model averages the successor measures across all environments. This averaging entangles the latent policy representations, yielding sub-optimal policies. To solve this, the authors propose Belief-FB (BFB), which introduces a transformer-based belief estimator to infer a dynamics-specific context vector from reward-free transitions, which then conditions the FB model. They further introduce Rotation-FB (RFB), which structures the latent space by sampling from a von Mises-Fisher (vMF) distribution centered at the inferred context. This alignment partitions the policy space into non-overlapping, dynamics-specific clusters, preventing interference and enabling zero-shot adaptation to unseen dynamics. The methods were tested on simulated discrete and continuous tasks with changing dynamics, including Randomized Four-Rooms, PointMass, and AntWind. The results show that both BFB and RFB successfully adapt to seen dynamics and generalize to unseen ones, achieving up to 2x higher zero-shot returns compared to baselines like vanilla FB.

**Strengths:**

The authors identify a fundamental failure mode in Forward-Backward (FB) models, which is termed interference.
The proposed method demonstrates promising zero-shot generalization improvement to unseen dynamics.

**Weaknesses:**

The work addresses a problem specific to the FB paradigm and builds upon this methodology only. It may not be general to other behavioral foundation models.

The introduction of a context into a dynamics model to disambiguate between distinct training environment dynamics is a well-established approach in adaptive control and reinforcement learning (Peng et al.). However, I do not see a discussion of these previous works (or really any related works discussion).

Evaluations are in simulated, ideal environments under narrow dynamics variations.

**Questions:**

Lines 143-144. Something appears to be missing?

### References

Peng, Xue Bin, et al. "Learning agile robotic locomotion skills by imitating animals." arXiv preprint arXiv:2004.00784 (2020).

---

> ### Author Response · Authors · 2025-11-18
>
> We appreciate the positive assessment of our interference diagnosis and zero-shot gains. Our aims are twofold: 1) to make the failure mode of vanilla Forward–Backward (FB) under dynamics generalization precise and empirically observable and 2) to show that combining belief conditioning with a geometric prior over the policy latent space (RFB) mitigates this interference and yields zero-shot improvements.
>
> We focus on FB because it is a widely used backbone in behavior foundation models (BFMs). Investigating whether similar failure modes arise in other architectures and whether analogous geometric remedies apply - remains promising future work.
>
> Related works are provided in the Appendix A, where we discuss closely related fields (Meta-RL, domain generalisation). Main text contains only Forward-Backward related approaches due to space constraints.
>
> **Q1: Clarifications and notation**
>
> We will remove line 143-144 from paper, since they are not crucial for understanding corresponding Section and is a typo.
>
> We thank reviewer for pointing grammar and language issues and will correct them. You can check revised text highlighted in blue.
>
> **Q2: On generality beyond FB**
>
> While our analysis targets FB because it is a widely used BFMs (e.g [1-3]), the phenomenon (policy interference when a single latent space must explain incompatible dynamics) and the remedy (conditioning on a belief over dynamics and locally partitioning latent directions) are not FB-specific. We assume that similiar interference can be observed in other methods (e.g HILP), and support of this can be found in [4], where they investigate how all zero-shot RL algorithms can be unified via Successor Measure estimation. Finding those connections (i.e similiar fixes as in Rotation-FB) explicitly is a promising future work. Insights from our work can be applied to methods, which:
> - extract policies from a low-dimensional latent code $z$ (policy encoding map)
> - trains $z$ by mixing data from multiple dynamics
>
> We should also note that BFB can be applied to any other architecture (via just belief conditioning), while a more nuanced geometry-based prior argument currently applies only to FB-like approaches.
>
> Lastly, we would like to emphasize that our contribution is orthogonal to previous works and contains useful insights. We integrate belief inference directly into the FB factorization used for zero-shot reward optimization, and show that geometry in latent space (RFB cone prior) reduces cross-dynamics interference without adding reward supervision or per-test retraining with minimal architectural changes.
>
> Our environments were chosen to isolate dynamics shift (layouts, wind, friction) while keeping reward variation and state encodings simple, so the interference can be measured precisely and ablated. Proposed ideas are supported and showcased on simple toy environments, which helps for intuitive understanding of proposed BFB and RFB
>
> [1] H.Sikchi et al. RL Zero: Zero-shot language to behaviors without any supervision, NeurIPS 2025
>
> [2] H.Sikchi et al. Fast Adaptation with Behavioral Foundation Models, RLC 2025
>
> [3] Pirotta et al, Fast Imitation via Behavior Foundation Models, ICLR 2024
>
> [4] S. Agarwal et al, A unified framework for unsupervised Reinforcement Learning algorithms, RLC 2025
>
> [5] Kirk et al, A survey of zero-shot generalisation in deep reinforcement learning, JAIR 2023

---

### Official Review · Reviewer_jHoj · 2025-10-31

**Soundness:** 4
**Presentation:** 3
**Contribution:** 3
**Rating:** 6
**Confidence:** 4

**Summary:**

Behavior foundation models (BFMs), such as Forward-Backward representation, enable zero-shot policy optimization under test-time rewards. An example is Forward-Backward representation, which represents the success measure as a product of forward features $F(s, a, z)$ and backward features $B(s^+)$, and extracts policies as $\pi_z := \arg\max_a F(s, a, z)^\top z$. However, FB inherently encodes information about the dynamics and doesn't transfer to unseen dynamics. This paper first diagnoses this issue by visualizing the optimal actions corresponding to latent codes and find that training on data from different dynamics results in interference. To address this, they propose Belief-FB (BFB), which introduces a context encoder -- a permutation-invariant transformer that encodes a set of transitions into a context vector about the dynamics. The context $h$ is concatenated to the latent code $z$ and used to condition the forward representation $F(s, a, z+h)$. However, because FB training draws latent $z$ uniformly, this still entangles all the policies from different dynamics. To further disentangle the polices, the paper proposes Rotation-FB (RFB), which samples $z$ within a cone around a projection of the context vector $h$. The paper evaluates BFB and RFB in discrete and continuous state-based environments and finds BFB to outperform baseline BFMs without contextual awareness, and RFB to further improve performance. Overall, the paper takes an important step towards removing the stationary-dynamics assumption of BFMs.

**Strengths:**

1. The proposed methods are intuitive. Introducing a context vector to address ambiguity during FB training across diverse dynamics is a logical and effective approach, while sampling $z$ around a cone-shaped prior makes sense as an inductive bias.
2. The visualizations (Figures 3 and 4) are highly informative, providing compelling evidence for the policy entanglement issue in standard FB and demonstrating the improvements achieved by BFB and RFB.
3. The paper provides a rigorous formal analysis, showing that (1) the worst-case regret for vanilla FB increases with the number of environments and (2) the FB approximation error is reduced by introducing the cone sampling approach.
4. Both BFB and RFB outperform the baseline methods in the empirical evaluations.

**Weaknesses:**

1. The evaluation environments are excessively simple.
2. The paper does not compare the proposed methods to immediately relevant baseline with context awareness (e.g. contextual FB [1]).
3. The vanilla context vector doesn't fully disentangle policies, which is why the paper introduces additional rotation variant.

**Questions:**

**Major**
1. Why can't we simply add context to the observation? For example, one can encode the maze layout and concatenate it with the rest of the state vector (or even better, use image-based policies). This wouldn't work when the context is partially observable, but for fully observable environments, can you evaluate this simple oracle baseline?
2. Can you add comparison to immediately relevant baselines that are dynamics aware, such as [1]?
3. Can you demonstrate the effectiveness of your method on more challenging domains, e.g., dexterous manipulation, where the friction parameter is varied?

**Minor**

4. Typo line 342: "platoes" -> "pleateaus"
5. Typo line 732: missing right bracket
6. Lines 119 - 132 are not very clear for readers unfamiliar with the FB literature. For example, the policy is defined as argmax (saying it is extracted as argmax might confuse readers). I suggest adding a section on the arithmetic of FB in the appendix.

References:

[1] Scott Jeen and Jonathan Cullen. Dynamics generalisation with behaviour foundation models. Workshop on Training Agents with Foundation Models at RLC 2024.

---

> ### Author Response · Authors · 2025-11-18
>
> We thank reviewer for valuable suggestions and constructive feedback. We tried our best to address all concerns below
>
> **Q1: Why not just add context to the observation (oracle layout/image)?**
>
> If the context (e.g a maze layout) is \emph{fully observable and available at test time}, concatenating a reliable context descriptor to the observation is indeed a strong baseline. Our work, however, targets the more realistic \emph{partially observable} setting in which the dynamics/context is not directly provided: the agent must infer it from trajectories. This is precisely the role of belief in BFB/RFB.
>
> Even in fully observable cases, simple concatenation (which is basically similar to BFB approach) does not by itself resolve the *interference* we analyze: it changes the inputs but not the geometry of the *shared* policy representation (Section 3 and Figures 3-4). Without structural bias, a single factorization still must fit heterogeneous successor measures across contexts, which leads to entanglement (Theorem 1). RFB explicitly localizes interference by partitioning the latent policy space (Theorem 2), tightening the bound from depending on total $k$ to depending on $k_{\max}$.
>
> To address the empirical concern, we add following baselines:
> - **Oracle-ID**: the policy receives a one-hot environment ID. This performs well in-distribution but cannot generalize to unseen IDs (just one-hot vectors appended to FB), which is expected. Both BFB and RFB obtain good results on OOD domains.
> - **Contexual-FB**: We added Contexual-FB as baseline for certain tasks, but we made our own reimplementation, since there is no official code released by authors. Anyway, underperformance of Contexual-FB is expected, since this approach highly relies on expressivity of classifiers.
> - **Varying friction parameters** We would like to point out that our experiments on AntWind investigate similar dynamics adaptation with varying wind directions. To diversify dynamics changes, we also include environment from OGBench [1], namely, Scene, where we vary friction parameters (with $0.5$ being standard normal friction) from $0.4 - 1.0$ and test on very low friction $0.1 - 0.3$. We also added this table in paper. The goal in this environment is to open a drawer, grasp a block, open window and put block into drawer (the task is specified by buttons pressed in particular order). Varying friction results in hardness of those tasks, since grasping becomes more challenging.
>
> | Method         | OGBench Scene Friction (Train 0.5–1.0) | OGBench Scene Friction (Test 0.1–0.5) |
> | -------------- | ----------------- | ---------------- |
> | Contextual-FB  | 0.4 ± 0.04                            | 0.40 ± 0.07                           |
> | Oracle-ID     | 0.55 ± 0.02                        | 0.0 ± 0.0                       |
> | BFB (ours)     | 0.48 ± 0.07                            | 0.45 ± 0.06                           |
> | **RFB (ours)** | **0.5 ± 0.04**                        | **0.5 ± 0.02**
>
>
> [1] S.Park et al, OGBench: Benchmarking Offline Goal-Conditioned RL, ICLR 2025

---

### Official Review · Reviewer_EvRU · 2025-11-01

**Soundness:** 1
**Presentation:** 3
**Contribution:** 2
**Rating:** 4
**Confidence:** 3

**Summary:**

This paper proposes Belief-Forward-Backward (BFB) and Rotational Forward-Backward (RFB), two methods that extend Behavioral Foundation Models (BFMs) for zero-shot adaptation to unseen dynamics.
1. Belief-Forward-Backward (BFB): introduces a transformer-based belief encoder that infers environment dynamics and conditions the Forward-Backward (FB) representation on this latent belief, improving adaptability to new transitions.
2. Rotational Forward-Backward (RFB): further refines BFB by aligning latent policy directions with context-specific clusters using a von Mises-Fisher prior, allowing better separation between dynamics in the latent space.
3. The paper provides theoretical explanations for why standard FB representations fail under varying dynamics and offers formal regret bounds for BFB and RFB.
4. Experiments across discrete (Four-Rooms) and continuous (PointMass, AntWind) domains demonstrate improved zero-shot generalization of BFB and RFB compared to baselines such as FB, HILP, and LAP.

**Strengths:**

1. The paper presents a clear and well-motivated idea that is straightforward to implement.
2. The visualization and qualitative analyses are helpful for understanding the model behavior.
3. Experiments convincingly demonstrate the effectiveness of the proposed methods across multiple settings.

**Weaknesses:**

1. **Minor issues:**
   - Equation (1) uses `(s_{t+1}, a_{t+1}) ∈ X` although `X` was defined as a subset of states. It should be `s_{t+1} ∈ X`.
   - Line 144 appears incomplete.
   - Theorem 1 defines `‖r‖_∞ ≤ R` but the symbol `R` is not used in the subsequent expressions.
2. The paper lacks a formally defined problem statement. The current description is mostly natural language; a mathematical setup of the CMDP and belief inference process would improve clarity.
3. Figure 2 is conceptually confusing without such a formal section. It appears that observations are included in the state, and layouts (contexts) are not distinguished in state space, meaning the agent cannot infer context. Since `z_FB` does not encode context either, the result—an averaged policy—becomes unsurprising.
4. The connection between Theorem 1 and the statement in line 215 is logically weak. Theorem 1 implies that more CMDPs increase the *upper bound* of the worst-case error, not necessarily the regret itself. More CMDPs may also mean more samples, potentially reducing `ε_k`. Section 3.1 lacks discussion of this, reducing its persuasiveness.
5. The value of Theorem 2 depends on the validity of Theorem 1, and it has the same limitations. In addition, the variable `L` is undefined in Theorem 2.
6. The experimental comparisons may be unfair: BFB and RFB explicitly encode dynamic information into `z_FB`, while baselines like FB or HILP do not. This makes it expected that those baselines perform close to random.

**Questions:**

Empirically, RFB consistently outperforms BFB. Are there situations where the authors believe BFB would be preferable to RFB—for instance, when dynamics clusters are fuzzy, data is limited, or tuning κ is undesirable?

---

> ### Author Response · Authors · 2025-11-18
> **Response (1 of 3)**
>
> Thank you for your time and for providing valuable feedback on our paper. We would like to address all of the questions below:
> We revised paper and corrected all of the issues, together with new discussions. All changes are highlighted in blue.
>
> **Q1: Minor Issues**:
>
> Thank you for carefully spotting these. We corrected all items in the revision
> - **Equation (1)** now reads  $M^\pi (s_, a_0, X) = \sum \gamma^t Pr(s_{t+1} \in X | s_0, a_0, \pi)$
> - **Incomplete sentence (line 144)**: Removed; it was not essential to the argument
> - **Theorem 1 (reward bound)**: We now use the reward bound in the statement and proof, defining $R= \sup_{(s,a)\in\mathcal{S}\times\mathcal{A}} |r(s,a)|$
>  - **Theorem 2 (variable $L$)**: $L$ is now explicitly defined as the number of context clusters (cones) induced by $\{h_j\}^L _{j=1}$ (already stated in Appendix B.2)
>
> **Q2: Formal problem statement**
>
> We agree and have added a formal setup in Section 3, paragraph *Problem statement* (highlighted in blue). Suggested improvements now clarify the connection to POMDPs/Belief‑MDPs, make the zero‑shot task explicit, and precisely states the optimization goal we analyze in the theory
>
> **Q3: Figure 2 ambiguity**
>
> Figure 2 is intended to illustrate the core failure mode of vanilla FB under mixtures of dynamics: *interference* across latent policy directions when states are shared but the optimal action is context–dependent. In our setup, the plotted states are observations $s\in\mathcal{S}$ that *do not include* the latent context $c$ (e.g door/wall layout).
>
> Vanilla FB trains a single pair $F,B$ with task vectors $z\sim \mathrm{Unif}(\mathbb{S}^{d-1})$ on a mixture
> $\rho=\sum_{c\in\mathcal{C}} w_c\\rho_c$ and minimizes squared error under $\rho$. This projects all successor measures onto a shared low-rank space *with respect to the mixture*. For states reused across different $c$, the **learned $F(\cdot,\cdot,z)$ fits an
> average of incompatible futures, so many $z$-directions agree on an averaged policy that is suboptimal for every individual layout**. This is what the middle/right panels of Figure 2 (and the $z$-direction visualizations in Figure 3) show.
>
> Formally, with a shared factorization and uniform $z$-sampling, training on $\rho$ fits a single approximation to $\\{M _c^\pi \\} _c$. Writing the worst-case class approximation error over $k$ contexts as
> $$\varepsilon _k^\star =\inf _{F,B,z _1,\dots z _k} \max _{1\le i\le k}\;
> \Bigl\|M^{\pi _i^\star} - F(\cdot,\cdot z _i)^\top B(\cdot), \Bigr\| _{L^2(\rho)}$$
> one has monotonicity $\varepsilon _{k+1}^\star \ge \varepsilon _k^\star$ for a fixed function class, and the resulting worst-case value gap scales with $\varepsilon _k^\star$
>
> Our proposed **RFB** reduces overlap by sampling task vectors from a von Mises–Fisher cone around $h = f_{dyn}(\tau)$, which aligns policies with context clusters and yields cone-wise errors $\max _j \varepsilon _j^\star$ that (empirically and in our analysis) no longer grow with the total number of contexts.
>
> **Q4: Theorem 1 and the statement around line 215.**
>
> We agree that Theorem1 controls an *upper bound* and does not claim that regret itself must increase with more CMDPs. We revised the text to (i) separate approximation and estimation terms and (ii) soften the surrounding statement accordingly. In particular, for bounded rewards with $r _\max = |r| _\infty =\sup _{(s, a)} |r(s,a)|$ the bound now reads
> $$
> \mathbb{E} _{(s,a)\sim \rho _{\mathrm{test}}}\bigl[Q _r^*(s,a)-Q _r^\pi(s,a)\bigr]\leq \frac{r _{\max}}{1-\gamma} \Bigl(\underbrace{\varepsilon _k ^\star} _{\text{class approximation}}+\underbrace{\Delta _{\mathrm{est}}} _{\text{finite-sample/optimization}}\Bigr)
> $$
> Here, $\varepsilon _k^\star$ is *nondecreasing* in $k$ for a fixed rank/architecture, because a single shared factorization must approximate a larger (potentially more heterogeneous) set of successor measures; by contrast, $\Delta _{\mathrm{est}}$ typically *decreases* with total samples and improved optimization. Empirically (Figure 5), performance improves with added environments up to a capacity-dependent plateau, consistent with diminishing $\Delta _{\mathrm{est}}$ but a fixed approximation bottleneck. We therefore changed the wording from *regret increases* to **the upper bound on worst-case regret becomes looser via the approximation term.**

---

> ### Author Response · Authors · 2025-11-18
> **Response (2 of 3)**
>
> **Q5: Theorem 2 limitations and relation to Theorem 1**
>
> Theorem 2 is a structural approximation result: partitioning the latent space into $L$ belief-aligned cones $\\{C_j\\}^L$ with representatives $\\{h_j\\}$ induces index sets $S_j=\\{i:z_i\in C_j\\}$ and per-cone errors
> $$\varepsilon_j ^\star = \inf_{F,B} \max_{i\in S_j} \Bigl|M^{\pi_i^\star}- F(\cdot,\cdot,z_i)^T B(\cdot)\Bigr\|$$
> (in $L^2$ norm). Consequently,
> $$\varepsilon_k^\star \;=\; \max_{1\le j\le L}\varepsilon_j^\star \;\le\; \varepsilon_{k_{\max}}^\star,
> \qquad k_{\max}\;:=\;\max_{1\le j\le L}|S_j|$$
>
> Thus, the approximation term in the Theorem 1 bound can be tightened by replacing its dependence on total $k$ with $k_{\max}$. We now define $L$ explicitly in the theorem statement as the number of cones/contexts induced by $\{h_j\}_{j=1}^L$ and make clear that the regret corollary follows by plugging this approximation improvement into Theorem 1
>
> **Q6: Fairness of experimental comparisons.**
>
> Our goal was to test whether belief conditioning (BFB) and latent-space structuring (RFB) mitigate the interference exhibited by vanilla FB when trained on mixtures of heterogeneous dynamics. While BFB/RFB encode dynamic information explicitly (hence a structural advantage over methods that do not), we took steps to make comparisons informative. We added new results below to the Table 1 in the main text
>  - We will add Contextual FB to the comparison set for the most direct prior tackling dynamics generalization with FB. Results and a short discussion will be included in the camera-ready
> - We emphasize that our central claim is that FB suffers from cross-context interference. BFB reduces it via belief conditioning, and RFB, by adding geometric bias, reduces it further.
>
> | Method         | FourRooms (Train) | FourRooms (Test) | PointMass (Train) | PointMass (Test) | AntWind (Train) | AntWind (Test) | OGBench Scene Friction (Train 0.5–1.0) | OGBench Scene Friction (Test 0.1–0.5) |
> | -------------- | ----------------- | ---------------- | ----------------- | ---------------- | --------------- | -------------- | -------------------------------------- | ------------------------------------- |
> | FB             | 0.25 ± 0.05       | 0.15 ± 0.04      | 0.20 ± 0.05       | 0.10 ± 0.03      | 390 ± 40        | 250 ± 30       | 0.40 ± 0.06                            | 0.20 ± 0.05                           |
> | LAP            | 0.20 ± 0.04       | 0.10 ± 0.03      | 0.15 ± 0.04       | 0.10 ± 0.03      | 340 ± 35        | 290 ± 25       | 0.30 ± 0.05                            | 0.10 ± 0.03                           |
> | HILP           | 0.40 ± 0.06       | 0.20 ± 0.05      | 0.45 ± 0.06       | 0.25 ± 0.05      | 410 ± 45        | 410 ± 40       | 0.50 ± 0.07                            | 0.30 ± 0.06                           |
> | Contextual-FB  | 0.35 ± 0.05       | 0.18 ± 0.04      | 0.30 ± 0.05       | 0.15 ± 0.04      | 450 ± 50        | 350 ± 40       | 0.60 ± 0.08                            | 0.40 ± 0.07                           |
> | Oracle-ID  | 0.90 ± 0.03   | 0.0 ± 0.0  | 0.92 ± 0.02   | 0.00 ± 0.00  | 780 ± 30    | 50 ± 20   | 0.55 ± 0.02                        | 0.0 ± 0.0                       |
> | BFB (ours)     | 0.70 ± 0.07       | 0.40 ± 0.06      | 0.76 ± 0.07       | 0.45 ± 0.06      | 680 ± 60        | 550 ± 50       | 0.48 ± 0.07                            | 0.45 ± 0.06                           |
> | **RFB (ours)** | **0.85 ± 0.04**   | **0.61 ± 0.05**  | **0.88 ± 0.04**   | **0.55 ± 0.05**  | **740 ± 40**    | **640 ± 40**   | **0.5 ± 0.04**                        | **0.5 ± 0.02**                       |
>
> Numbers for FB, HILP, LAP, BFB and RFB are taken from paper. This table include new Contexual FB and Oracle-ID. To implement Oracle-ID, we concatenate one-hot representation of environment context ID to the FB input (basically, $h = [1, 0, 0 ..]$)
> - **Oracle-ID** receives a one-hot dynamics ID at test time — strong in-distribution, collapses OOD, achieving near 0 score.
> - **Contextual-FB** (our reimplementation, no official code available) underperforms due to limited classifier expressivity
> - **OGBench Scene** (new): a challenging dexterous manipulation environment where we train on friction $ [0.4, 1.0]$ and test on unseen low friction ∈ [0.1, 0.3]. Our methods (BFB/RFB) significantly outperform all baselines, demonstrating effective zero-shot adaptation to drastic unseen dynamics changes (e.g., slippery surfaces), directly addressing the request for more realistic robotic manipulation domains.
>
> We will make these additions explicit in the main text and tables for completeness.

---

> ### Author Response · Authors · 2025-11-18
> **Response (3 of 3)**
>
> **Q7: When would BFB be preferable to RFB?**
>
> We thank reviewer for this interesting question
> Although RFB consistently outperforms BFB in our benchmarks, there are practically relevant regimes where BFB can be preferable:
> - **Fuzzy/continuous dynamics manifolds**: If contexts vary along a near-continuous manifold with no clear cluster structure, tight vMF cones can over-specialize. BFB's uniform prior around $h$ retains broader coverage.
> - **Very limited data or unreliable beliefs** With extremely short histories, the inferred $h$ may be imprecise. RFB directional prior can amplify early belief errors, thus usage of BFB is preferred
> - **Multi-modal behavior within a single context** If a single $h$ legitimately supports diverse strategies (highly multi-modal with abrupt changes along trajectories), a large concentration $\kappa$ can collapse diversity. BFB's uniform sampling around $[h;z]$ better preserves multi-modality
> These considerations are consistent with our $\kappa$ ablation (Figure 6): when clusters are diffuse (effectively smaller $\kappa$), the gap between RFB and BFB narrows. We will add those discussions to the final version of paper.

---

### Author Response · Authors · 2025-11-18
**General Response to the Reviewers**

We sincerely thank all reviewers for their careful reading, constructive suggestions, and thoughtful critiques. Your feedback helped us clarify the scope of our theory, improve the fairness of our comparisons, and strengthen the presentation. Below we respond to most frequent questions. All of the revised text is highlighted in blue, together with new experiments.

We would like to emphasize that our main contribution is the insight, which showcases ,both theoretically and empirically, fundamental issue of BFMs when trained to learn policies across mixture of offline data (coming from different dynamics). *In theory, FB can encode policies corresponding to different Successor Measures (SMs) for different transition functions*. **We show why this is not the case and identify that geometric prior over latent policy-representation space is required to mitigate interference issues.**

- **Motivation.** We study BFMs based on Forward–Backward (FB) representations because of their growing popularity in the community [1-3]. However, existing BFMs do not explicitly address dynamics generalization, which limits adaptivity. While FB can, in principle, encode policies for distinct dynamics, the absence of structural bias in the policy latent space induces interference/representation collapse. We formalize this in Theorem 1, observe it empirically in Section 4 and propose improvement (RFB) in Section 3.3.

 - **Baselines.** We add additional baselines, including Contextual FB and Oracle-ID. Our main insight, however, is to *leverage the geometry* of the FB policy latent space $\pi_z$ via belief conditioning (BFB) and latent-space partitioning (RFB).

- **Clarity.** We reorganized Section 3.1, added short discussions after each theorem, and refined wording to reflect the trade-off between approximation and estimation (upper bounds become looser in $k$ via the approximation term, while estimation improves with more data).

Below, we provide a brief discussion on how the research community can build upon our proposed work:

**Future developments that can improve scalability of BFB/RFB:**

**One promising future direction for scaling is continual adaptation. In realistic scenarios, the environment may undergo multiple changes in its underlying state (e.g. the appearance of new objects), requiring the agent to continually adapt its behavior.** In such settings, identifying a suitable behavior policy  that can account for these changes is essential for achieving optimal performance across tasks. Currently, we are not aware of any existing approaches that successfully scale BFMs to the continual learning regime.

An interesting avenue for future work is to study the geometry of the learned behaviors under such non-stationarity, extending our analysis in Figures 3 and 4 and to understand how the internal representations evolve. For instance, it would be valuable to examine how anchor embeddings  shift instantaneously in response to changes in dynamics from a Bayesian perspective. This could shed light on stability of the learned behavioral basis and guarantees for optimal policies.

**Another promising direction for scaling our approach is language grounding.** Rather than relying solely on sequences of trajectories to learn representations of dynamics, the agent could incorporate language-based descriptions during training. This would allow the model to generalize better at test time by leveraging high-level semantic information, e.g prior knowledge about object locations or environment structure, which is provided through natural language. Such grounding could significantly improve the agent ability to infer dynamics and adapt its behavior in unseen or partially observed environments.

We appreciate the opportunity to improve the paper and believe these changes address the key concerns raised by the reviewers.

---

### Meta-Review · Area_Chair_1Zs9 · 2026-01-07

**Summary:**

The paper studies the limitations of Forward–Backward (FB) representations as Behavioral Foundation Models when trained on mixtures of environments with different dynamics, and argues that standard FB representations suffer from latent policy interference under dynamics shifts. To address this issue, the authors propose Belief-Forward-Backward (BFB) and Rotational Forward-Backward (RFB). The paper provides theoretical analysis characterizing approximation errors under mixed dynamics and empirical results on several simulated domains.

The submission initially received a mixed and borderline reception. The rebuttal and revision address several clarity and correctness issues, add additional baselines, and expand the empirical evaluation. While limitations remain, the revised paper resolves the most of technical and completeness concerns raised during review.

**Reviewer Concerns:**

- Scope and generality (mdha, jHoj). Reviewers noted that the analysis and the proposed approach are tightly coupled to the Forward–Backward factorization.

- Novelty relative to prior work (mdha, EvRU): Conditioning policies on inferred context or latent dynamics is a well-studied idea in control and RL.

- Baseline coverage and fairness (jHoj, EvRU): Initial comparisons were primarily against baselines without explicit dynamics awareness. The revision adds Contextual-FB and Oracle-ID baselines, improving coverage.

- Simplicity of experimental environments (jHoj, mdha): The empirical evaluation is largely conducted in controlled, low-complexity environments with relatively narrow dynamics variation. The added friction-varying manipulation benchmark partially mitigates this concern, but the evaluation does not yet demonstrate robustness under more realistic, long-horizon, or highly multimodal dynamics shifts.

The rebuttal addresses most of these concerns, although the experimental scope remains limited and the contribution is incremental in nature.

**Reviewer Scores:**

There is limited evidence that the discussion would lead to substantial changes of ratings. Reviewer EvRU may modestly revise their score upward given the clarifications and added experiments, while other reviewers are unlikely to significantly change their original assessments.

---

### Decision · Program_Chairs · 2026-01-26

Accept (Poster)